# Rising rainfall intensity induces spatially divergent hydrological changes within a large river basin

Yiping Wu [1,2], Xiaowei Yin[1], Guoyi Zhou [3] ✉, L. Adrian Bruijnzeel[4,5], Aiguo Dai [6], Fan Wang[1], Pierre Gentine [7], Guangchuang Zhang[1], Yanni Song[8] & Decheng Zhou [3]

Droughts or floods are usually attributed to precipitation deficits or surpluses, both of which may become more frequent and severe under continued global warming. Concurring large-scale droughts in the Southwest and flooding in the Southeast of China in recent decades have attracted considerable attention, but their causes and interrelations are not well understood. Here, we examine spatiotemporal changes in hydrometeorological variables and investigate the mechanism underlying contrasting soil dryness/wetness patterns over a 54-year period (1965–2018) across a representative mega-watershed in South China—the West River Basin. We demonstrate that increasing rainfall intensity leads to severe drying upstream with decreases in soil water storage, water yield, and baseflow, versus increases therein downstream. Our study highlights a simultaneous occurrence of increased drought and flooding risks due to contrasting interactions between rainfall intensification and topography across the river basin, implying increasingly vulnerable water and food security under continued climate change.

Drought is a periodic natural phenomenon that may last for weeks to years; initially typically resulting from precipitation deficit, drought generally leads to reduced water availability and plant growth[1–7]. In contrast to the permanent aridity of dry regions, droughts can occur in water-rich areas as well, reflecting anomalous fluctuations in local or regional precipitation. Severe droughts have affected nearly every continent in recent decades[8,9]. Flooding or excessive wetness, is usually caused by prolonged and intense rainfall, and may be exacerbated by widespread land degradation and urbanization or by snowmelt in cold regions[10–13]. Floods may occur suddenly or gradually but often have severe impacts on agriculture, economy, and society at large[14–16].

Both droughts and floods are common and widespread, and form the costliest meteorological disasters[8,9]. Between 1984 and 2017, droughts caused an estimated average annual loss of USD 16.5 billion globally, accounting for 13% of the total economic costs arising from meteorological disasters; for China, this fraction reached 20% with an average annual loss of USD 6.3 billion (44.4 billion RMB)[17]. Moreover, drought-induced tree mortality and associated reductions in ecosystem services have been reported around the globe[18–20]: in Europe,

[1]Institute of Global Environmental Change, Department of Earth & Environmental Science, Xi'an Jiaotong University, Xi'an, Shaanxi 710049, PR China. [2]National Observation and Research Station of Regional Ecological Environment Change and Comprehensive Management in the Guanzhong Plain, Xi'an 710061, PR China. [3]Institute of Ecology, School of Applied Meteorology, Nanjing University of Information Science and Technology, Nanjing 210044, PR China. [4]Department of Geography, King's College London, London WC2B 4BG, UK. [5]Institute of International Rivers and Eco-Security, Yunnan University, Kunming 650091, PR China. [6]Department of Atmospheric and Environmental Sciences, University at Albany, State University of New York, Albany, NY 12222, USA. [7]Department of Earth and Environmental Engineering, Earth Institute, Columbia University, New York, NY 10027, USA. [8]Technology Innovation Center for Land Engineering and Human Settlements, Shaanxi Land Engineering Construction Group Co. Ltd and Xi'an Jiaotong University, Xi'an 710115, PR China. ✉e-mail: gyzhou@nuist.edu.cn

droughts killed ~500,000 ha of forest between 1987 and 2016[21,22]; the severe drought of 2010 in Amazonia temporarily turned the forest from a carbon sink to a net source[23,24]; and in 2013, a one-in-a-century drought in South China caused a reduction in carbon sequestration of 101.5 Tg C[25]. Similarly, flooding accounted for 34% of all natural disasters registered world-wide between 1960 and 2014 (on average 17 major floods/year)[26], causing a total damage of USD 651 billion globally between 2000 and 2019[27] and increasing the percentage of global population exposed to floods by ~20% between 2000 and 2015[28]. In China, ~11.42 million hectares of cropland were affected annually by flooding between 1978 and 2018, representing an average annual loss of 187.32 million tons of grains[29].

Given the rapid and successive warming and altered rainfall regimes observed in recent years, climate change may well increase the frequency, extent, and severity of droughts and floods in future, thereby putting securities of water, food, and ecological systems at greater risk[3,28,30–34]. Globally, dry areas have increased from 20% in 1950–1979 to over 30% after the 1990s[35]. The frequency of nearly all grades of drought (from moderate to exceptional) has been projected to increase substantially (by 17–34%) over most continents by the late 21st century[36]. Similarly, flood frequency and duration have increased globally between 1985 and 2015[37]. Overall, rising climatic extremes imply a growing vulnerability of ecological and socio-economic systems under continued climate change, which has sparked substantial scientific interest[38–44]. However, although it is generally thought that droughts and floods may both become more severe (both in degree and extent) due to intensification of the hydrological cycle under continued global warming[45], predicting the associated changes in frequency, duration, and severity for specific regions is challenging because of the often large spatiotemporal heterogeneity of land surface conditions[46–49]. In particular, the mechanism of how spatiotemporal changes in rainfall intensity within large river basins affect patterns of soil drying or wetting and their associated hydrological responses at different levels of scale has not been examined sufficiently.

China has long suffered from extended and severe droughts that often led to major environmental and socio-economic losses[17,50]. Southwest China, in particular, has experienced extreme and sustained drought conditions since the beginning of the 21st century, with devastating consequences for agriculture and natural ecosystems[50–52]. For example, the 2010 spring drought caused a 20% reduction in hydropower production, a shortage of drinking water for 60 million people, and USD 3.5 billion worth of economic losses[53–55]. Accompanying the droughts in the Southwestern parts of the country, floods have been frequently reported for Southeast China[56–58]. Both the annual peak discharge and the number of extreme flood events in the West River Basin (the largest tributary of the Pearl River) have been increasing in recent decades[59–61]. In the same river basin, an extreme (200-year return period) flood occurring in June of 2005 affected over 4 million hectares of land, causing an economic loss of nearly USD 2 million (13.6 billion RMB), 131 casualties, and rendering 12.63 million people homeless[62]. Pertinently, both droughts in the Southwest and flooding in the Southeast occurred simultaneously in some years[47,63]. Several studies examined the recent droughts and floods in the region[41,46,48,49], assessing their spatiotemporal patterns and emphasizing the potential role of El Niño/Southern Oscillation (ENSO)[44,49,64]. However, it is still unclear whether, and to what extent, the drying in the Southwest and the flooding in the Southeast are related. In particular, it is not well understood whether the two phenomena can be attributed to a single (e.g., altering precipitation patterns) or multiple climate factors. These questions motivated us to examine how precipitation and other hydroclimatic fields have changed in the past decades, what the resulting hydrological responses have been, and to what extent regional drying and flooding are potentially interrelated.

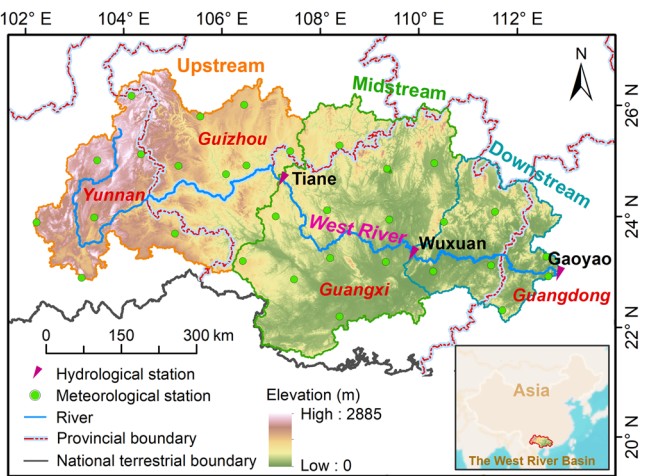

**Fig. 1 | Location of the West River Basin (WRB) within South China plus locations of climatic and stream gauging stations used in the present analysis.** The green dots mark the meteorological stations. The red triangles mark three streamflow gauging stations. The inset highlights the location of the WRB with an East Asian monsoon climate.

For this purpose, we selected the 353,120 km² West River Basin (WRB) because it spans the Southwestern (headwaters) and Southeastern (downstream lowland) parts of the China (Fig. 1). In particular, we aim to investigate: (i) the spatiotemporal patterns of changes in precipitation and other hydrometeorological variables between 1965 and 2018; (ii) the associated changes in hydrological responses; and (iii) the primary evolutionary mechanism of dry and wet spells (drought/flood risks) across the basin and their possible interrelation. We hypothesize that rainfall intensification constitutes the primary driver of both drought occurrence (in the Southwest) and flooding (in the Southeast).

## Results

### Trends of rainfall intensity and other major climate variables

To examine how the climate across the WRB changed during the 54-year study period (1965–2018), we investigated the spatial patterns and temporal changes of rainfall and several other major climate variables, notably air temperature (maximum and minimum values), wind speed, relative humidity, and solar radiation using daily observations from 31 national meteorological stations distributed across the basin (Fig. 1).

We first analyzed the change of rainfall intensity. As shown in Fig. 2a, 26 out of the 31 stations showed an upward trend for the amount of rainfall per rain day (rainfall intensity index, RI, in mm d⁻¹; see Methods) with a locally observed maximum annual rate of 0.04 mm d⁻¹ y⁻¹. Eight stations (two in the upstream and three in the mid- and downstream, respectively) exhibited statistically significant rising trends for RI, despite a slight downward trend for annual rainfall in the upstream region and non-significant trends for the mid- and downstream regions (Supplementary Fig. 1a). The slope of the upward trend for RI steepened in the downstream direction (i.e., from 0.01 mm d⁻¹ y⁻¹ in the upper part to 0.03 mm d⁻¹ y⁻¹ in the downstream part; Fig. 3a–c), indicating an increase in the intensification of rainfall from the headwaters towards the basin outlet. Amounts of light rain (events <10 mm d⁻¹) decreased at all stations, with average trends of −0.83, −0.76, and −0.54 mm y⁻¹ for the upstream, middle, and downstream parts of the basin ($P < 0.01$; Fig. 3d–f). This basin-wide decrease in light rains implies an increased proportion of non-light rains (i.e., increased rainfall intensity) considering the stable annual amounts of rainfall in the mid- to downstream sections. Similarly, all stations showed a downward trend for the number of days with light rain (statistically

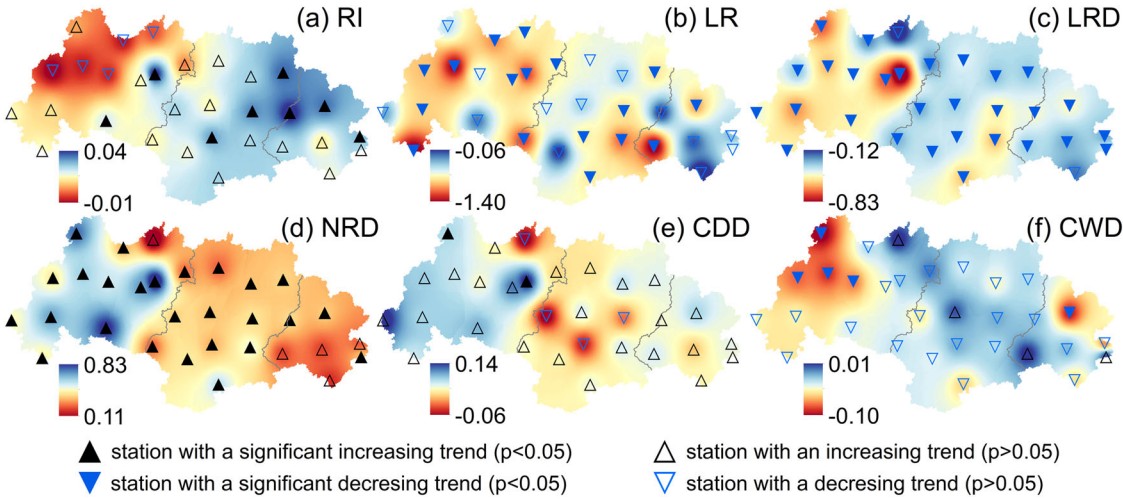

**Fig. 2 | Spatial distribution of annual trends of rainfall intensity-related factors in the West River Basin over the 54-year (1965–2018) study period. a** Rainfall intensity index (RI, mm d⁻¹). **b** Light-rain (<10 mm d⁻¹) amount (LR, mm). **c** Number of days with light rain (LRD). **d** Days without rain (NRD). **e** The maximum number of consecutive dry days (CDD). **f** The maximum number of consecutive wet days (CWD). Source data are provided as a Source Data file.

significant for 29 stations or 94%; Fig. 2c). Averaged per region, decreases in the number of days with light rain varied from −0.46d y⁻¹ in the upstream part to −0.33d y⁻¹ in the downstream part (Fig. 3g–i). Further, the number of days without rain increased significantly at 26 stations (84%), with significant average trends per sub-region of 0.55d y⁻¹ in the upstream part versus 0.31d y⁻¹ in the downstream part (Fig. 2d and Fig. 3j–l), indicating greater drying in the upstream direction. The longer dry intervals between rainfall events and the greater degree of drying in the upstream area are also reflected in the respective increases in the maximum number of consecutive dry days (Fig. 2e and Fig. 3m–o) and the reductions in the maximum number of consecutive wet days (Fig. 2f and Fig. 3p–r) across the basin.

Both maximum and minimum air temperatures presented overall upward trends throughout the basin, with a greater increase for minimum temperatures (up to 0.04 °C y⁻¹). However, the degree of warming did not show a clear spatial pattern (Supplementary Fig. 1b, c). For wind speeds, no overall temporal or spatial trends were found (Supplementary Fig. 1d). Similarly, no spatial patterns were found for relative humidity and solar radiation despite generally downward trends (Supplementary Fig. 1e–f). Therefore, the primary feature of the climate shift is a significant and substantial intensification of rainfall across the basin.

## Changes in key hydrological components
To examine the WRB's hydrological responses to its shifting climate— notably intensifying rainfall and longer dry spells—we calibrated/validated the Soil and Water Assessment Tool (SWAT). This process-based, spatially distributed hydrological model allows the derivation of spatiotemporal changes in hydrological components (see Methods). In this study, the key components simulated by SWAT includes soil water content, water yield, quick-response surface runoff (overland flow), and slow-response baseflow. We used linear regression to detect annual trends for the respective hydrological variables during the 54-year study period, with graphical representation at subbasin and hydrological response unit (HRU) levels as shown in Fig. 4.

Soil water content decreased significantly in the uppermost part of the WRB (at rates up to −2.0 mm y⁻¹ at HRU level), whereas the mid- and downstream parts generally showed increases (up to 2.0 mm y⁻¹ for some HRUs; Fig. 4a, b). Similarly, upstream water yields exhibited substantial decreases (up to −4.8 mm y⁻¹), accompanied by mostly modest increases in water yield lower down in the basin (although locally up to 4.2 mm y⁻¹; Fig. 4c, d). For surface runoff, a primary

contributor to flooding due to its typically quick response to rainfall, a similar spatial pattern emerged of significant decreases in some upstream areas (as high as −3.9 mm y⁻¹ locally) versus (often non-significant) increases further downstream (< 5.0 mm y⁻¹; Fig. 4e, f). Conversely, baseflow generally declined throughout the WRB, with the declines being most pronounced in the upstream parts (up to −3.2 mm y⁻¹; Fig. 4g, h). As such, the hydrological changes induced by climate change confirm the general pattern of drying in the upstream part of the basin versus a general wetting downstream. In addition, we found that slopes in the upstream area were generally higher than that in the mid- and downstream areas (Fig. 4i, j), which is generally consistent with the soil dryness/wetness patterns especially at the HRU level.

## Spatiotemporal patterns of dryness and wetness
To examine the spatiotemporal evolution of droughts in the WRB, we employed our previously-developed drought evaluation system, which integrated the hydrological modeling of SWAT and a water budget-based drought index, to derive the spatiotemporal series of the Palmer Drought Severity Index (PDSI; Methods) at a monthly time step. Next, we used linear regression analysis to detect annual trends in PDSI during the 54-year study period, with graphical representation at subbasin and HRU levels (Fig. 5). Annual values of PDSI decreased substantially, especially in the upstream part of the basin, at rates up to −0.11 y⁻¹ in some HRUs and up to −0.04 y⁻¹ in some subbasins (statistically significant). In contrast, PDSI values increased moderately in the downstream part, with maximum rates up to 0.03 y⁻¹ in some HRUs and up to 0.02 y⁻¹ in some subbasins (statistically non-significant; Fig. 5a, b). The spatial pattern again suggests strong drying in the upstream part, with the trend gradually becoming non-significant in the central part of the basin, and reversing to moderate wetting in the downstream area. We further identified areas as drying or wetting based on the spatial distribution of PDSI trends at HRU level. Considering the uncertainties of the modeling, trend values between −0.01 to +0.01 (instead of zero) were taken to be neutral. As shown in the pie chart of Fig. 5c, about 29% of the HRUs (mostly located in the upstream part) were drying and only 7% (mostly located in the mid- and downstream) were wetting up. This result aligns with the spatial hydrological response patterns identified in the previous section, indicating drought conditions were exacerbated mostly in the upstream part of the basin, both in terms of areal extent and degree (Fig. 5a–c). On a more limited scale, the lower

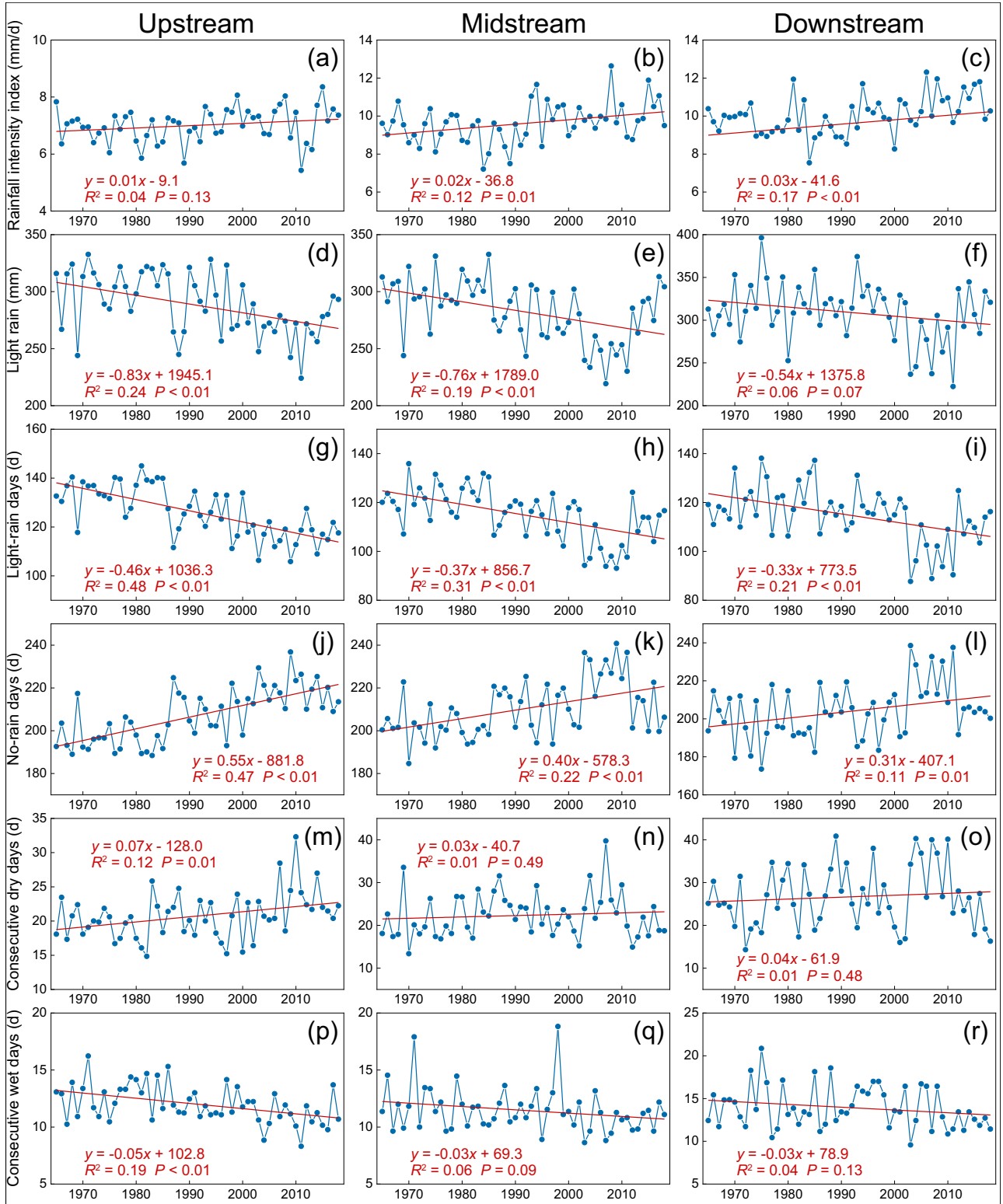

**Fig. 3 | Annual trends of six rainfall-related factors between 1965 and 2018 in the three sub-regions (up-, mid-, and downstream) of the West River Basin. a–c** Rainfall intensity index. **d–f** Light rain (<10 mm d⁻¹) amount. **g–i** Number of days with light rain. **j–l** Number of days without rain. **m–o** The maximum number of consecutive dry days. **p–r** The maximum number of consecutive wet days. Source data are provided as a Source Data file.

parts of the basin appear to become more liable to flooding (Fig. 4e, f and Fig. 5).

To further explore the temporal dynamics of the above spatial patterns of drying and wetting, we conducted an Empirical Orthogonal Function (EOF) analysis of the PDSI time series at the subbasin level

(Methods). The eigenvalues of five modes, together with explained variances and error ranges, are listed in Supplementary Table 1. According to the North significance test[65], the first three modes were statistically significant whereas the error range of the fourth mode overlapped with the fifth. Further, because the cumulative explained

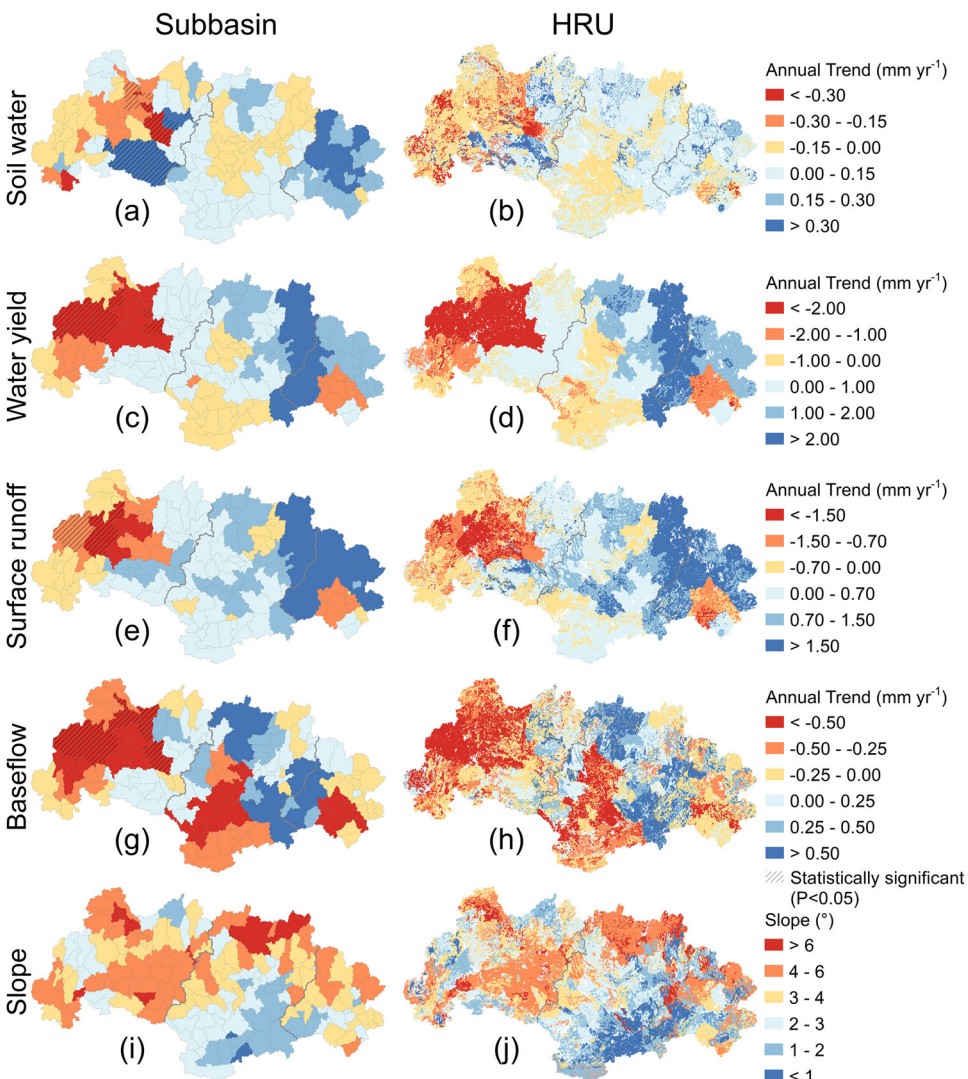

**Fig. 4 | Spatial distribution of annual trends for four hydrological components from 1965 to 2018 and slope across the West River Basin. a, c, e,** and **g** The annual trends for soil water content, water yield, surface runoff, and baseflow at subbasin level (shaded areas with statistically significant trends). **b, d, f** and **h** The annual trends for soil water content, water yield, surface runoff, and baseflow at hydrological response unit (HRU) level. **i, j** Slopes for subbasin and HRU level, respectively. Source data are provided as a Source Data file.

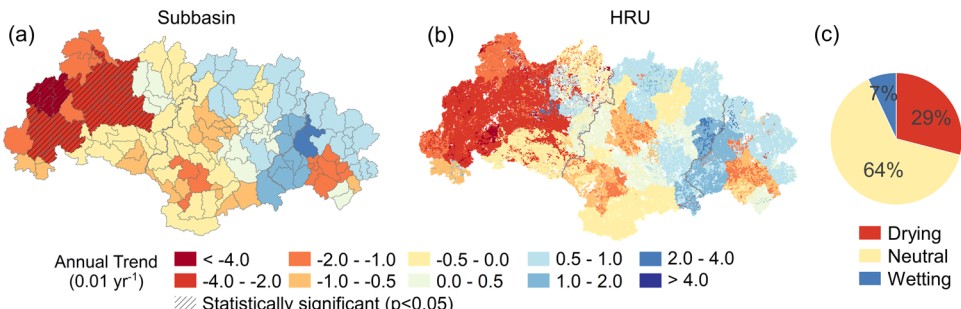

**Fig. 5 | Spatial distribution of annual trends of the Palmer Drought Severity Index (PDSI) in the West River Basin from 1965 to 2018. a** The annual trends for PDSI at subbasin level (shaded areas with statistically significant trends). **b** The annual trends for PDSI at hydrological response unit (HRU) level. **c** The pie chart summarizes the percentages of HRUs showing a drying, wetting, or neutral trend. Source data are provided as a Source Data file.

variance for the first two modes (49.6%) was not improved much by adding the third mode (Supplementary Table 1), we selected the first two leading modes to explain the main spatiotemporal features of PDSI. Figure 6a, b presents the spatial patterns of the two leading modes (EOFs), while their respective temporal variabilities (Principal Components, PCs) are shown in Fig. 6c, d, together with the results of the Mann-Kendall mutation test (Fig. 6e, f). The eigenvalues of EOF 1 were positive across all subbasins (Fig. 6a), indicating a consistent trend (either drying or wetting) across the entire basin. Further, positive or negative values of PC 1 (time coefficients) in a year indicated

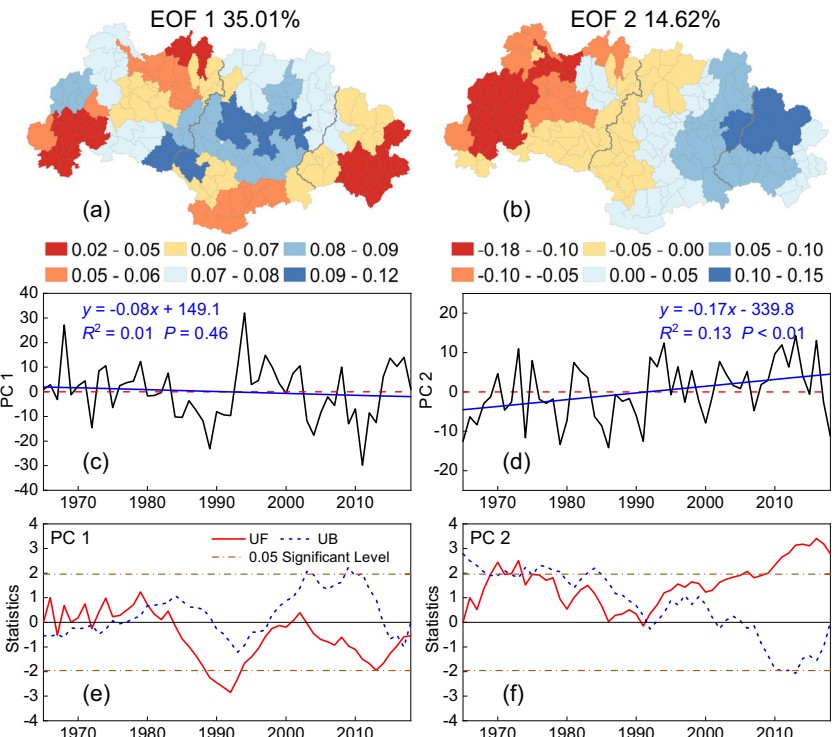

**Fig. 6 | Spatial patterns of the first two Empirical Orthogonal Function modes and time series of the principal components with mutation test for the Palmer Drought Severity Index at the subbasin level in the West River Basin between** 1965 and 2018. **a**, **b** Spatial patterns for the first two EOFs. **c**, **d** Time series of the principal components of the first two EOFs. **e**, **f** Mutation test for the time series of the principal components. Source data are provided as a Source Data file.

that the WRB, as a whole, was relatively wet or dry that year. As shown in Fig. 6c, wet and dry years either alternated or tended to be grouped. For EOF 2 (Fig. 6b), the eigenvalues were generally negative for the upstream part of the basin, whereas they became positive for the mid- to downstream parts, implying opposite trends for the PDSI in the respective regions. Further, a positive (negative) PC 2 in regions having a positive EOF 2 represented wetting (drying). According to Fig. 6d, most values of PC 2 were negative before 1992, suggesting the upstream part to be relatively wet and the downstream part relatively dry between 1965 and 1991. Values of PC 2 became positive from 1992 onward, indicating a reverse pattern of upstream drying and downstream wetting (Fig. 6d). Thus, the combination of EOF 2 and PC 2 explains the above opposite trends—a drying upstream and wetting downstream. The timing of this reversal was confirmed by the mutation test as the year 1992 (Fig. 6f). In addition, the new pattern of increased drying of the basin's headwaters and enhanced wetting of the lowlands became more pronounced after 2009 (i.e., when the UF curve exceeded the 0.05 significance level; Fig. 6f).

**Features of drought evolution in time and space**
Considering the mutation point for drought occurrence identified in Fig. 6f, we divided the 1965–2018 study period into two halves: 1965–1991 and 1992–2018. After quantitatively evaluating the degree and extent of drought in time and space (see Methods), we compared the differences between the two periods in terms of drought frequency, areal extent, duration, and severity.

Based on the time series of PDSI at HRU level for the entire period, the frequency of mild drought occurrence increased gradually in a downstream direction (up to 22% locally in the downstream) (Fig. 7a). A largely similar spatial pattern was identified for the first period (Fig. 7d), but no clear spatial gradient was observed for the second period (Fig. 7g). Overall, the spatial pattern for moderate droughts resembled that for mild droughts, although the magnitude of

frequency change was lower: generally <8% for the upstream part and <11% for the downstream part (Fig. 7b, e). In terms of severe drought occurrence, however, the up- and midstream areas experienced higher frequencies (by up to 16%) than the downstream area (whole period; Fig. 7c), with the gradual increase in the upstream direction becoming more pronounced during the second period (Fig. 7i). To further illustrate the trends in areal drought extent with time, the percentage areas suffering from the respective types of drought (mild, moderate, and severe) in the up-, mid-, and downstream parts of the WRB are presented in Supplementary Fig. 2. Areal extents of all three grades of drought increased with time in the upstream part, and the trends for mild (4.6% per decade) and moderate (4.2% per decade) droughts were statistically significant (Supplementary Fig. 2a, b). In the mid- and downstream parts, areal drought extent decreased slightly despite the non-significant trend.

Considering the entire study period, drought durations gradually increased in the downstream direction (Fig. 8a, b). Drought severity, however, showed a different spatial pattern with relatively higher values upstream and lower values in the mid- and downstream parts (Fig. 8c, d). Combined, the regional trends for drought duration suggested a significant upward trend in the upstream part (Supplementary Fig. 3a, b) and a non-significant downward trend downstream (Supplementary Fig. 3e, f). Considering the respective trends for the first (1965–1991) and second (1992–2018) period separately (Fig. 8e–l), there was a clear reversal in spatial pattern during the latter period, with generally longer drought duration and greater severity in the upstream part in recent decades (Fig. 8i–l). A similar change is apparent from the summary graphs of changes in drought duration and severity between the two periods, both at the subbasin (Fig. 9a) and regional levels (Fig. 9b). The cumulative duration of upstream droughts increased substantially (by 41.3%) from 63 months in the first period to 89 months in the second, whereas drought severity increased from 81.4 to 119.0 (an increase of 46.2%), indicating clearly

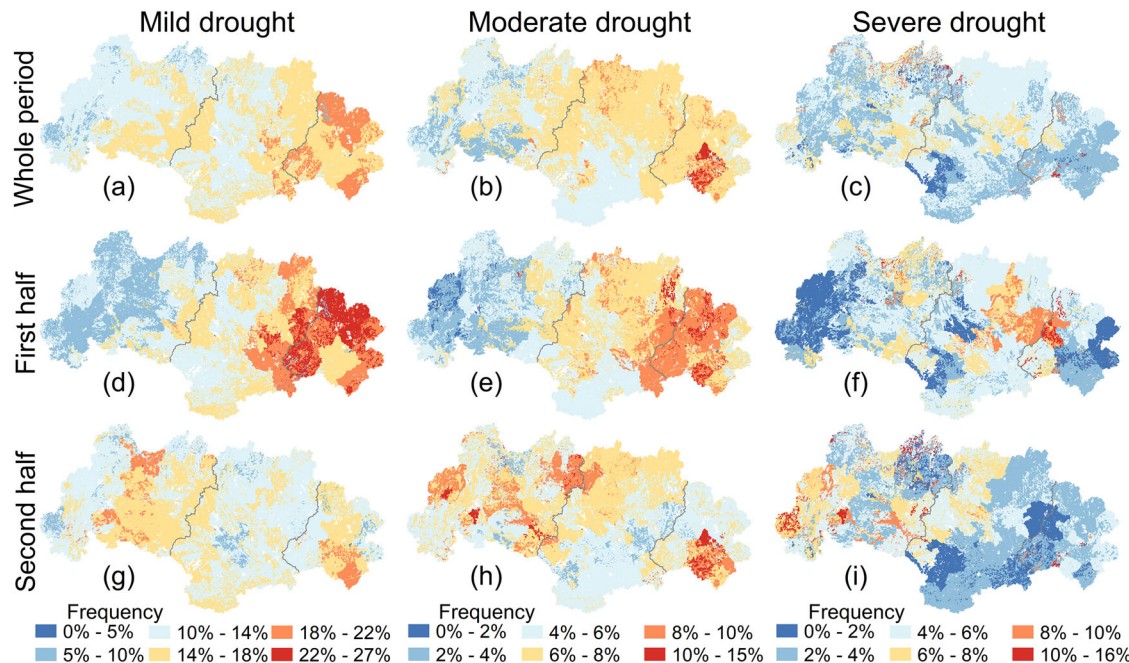

**Fig. 7 | Spatial distribution of frequencies of different grades of drought in the West River Basin at the hydrological response unit (HRU) level during the different periods. a–c** The frequencies of mild, moderate, and severe drought during the whole period (1965–2018). **d–f** The frequencies of mild, moderate, and severe drought during the first half (1965–1991). **g–i** The frequencies of mild, moderate, and severe drought during the second half (1992–2018). Source data are provided as a Source Data file.

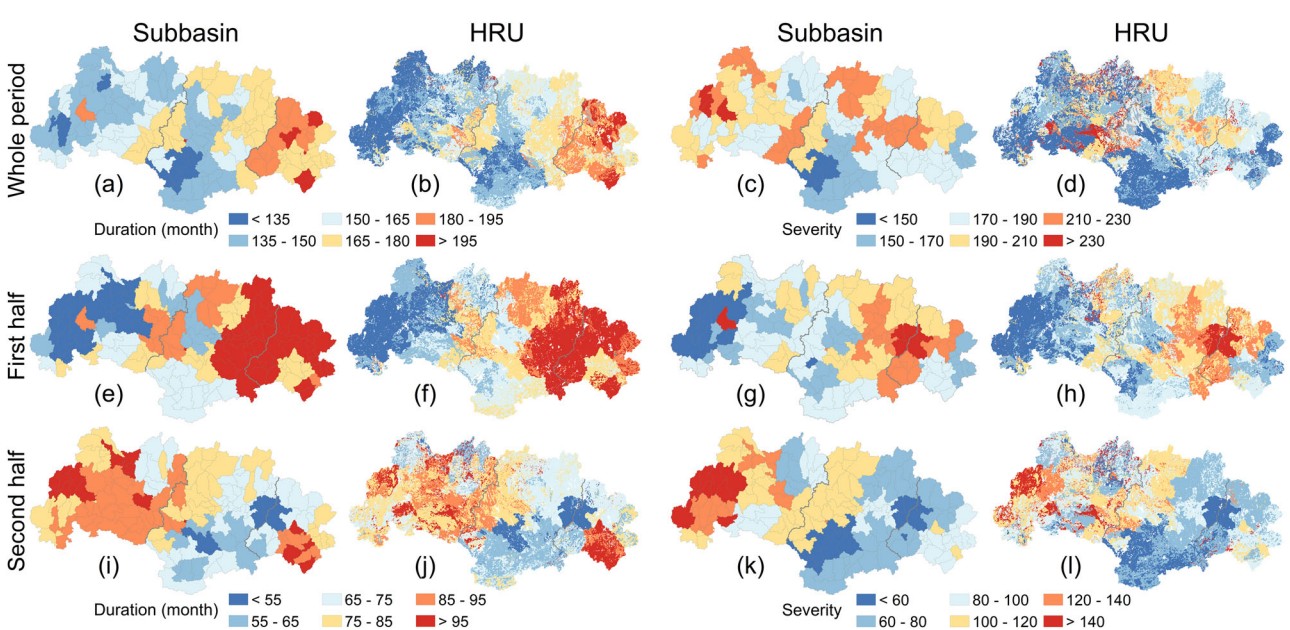

**Fig. 8 | Spatial distribution of drought duration and severity in the West River Basin during the different periods. a–d** The drought duration and severity at the subbasin and hydrological response unit (HRU) levels during the whole period (1965–2018). **e–h** The drought duration and severity at the subbasin and HRU levels during the first half (1965–1991). **i–l** The drought duration and severity at the subbasin and HRU levels during the second half (1992–2018). Source data are provided as a Source Data file.

exacerbated drought conditions. Conversely, drought duration in the mid- and downstream parts decreased by 17.4% and 22.9%, respectively, from the first to the second period. Similarly, drought severity decreased by 21.4% (midstream area) and 18.2% (downstream area), suggesting a shift to wetter conditions. Summarizing, the spatiotemporal pattern of changes in drought duration and severity confirmed the upstream drying and downstream wetting trends signaled earlier.

To further identify the most significant factors driving this spatially divergent pattern of drought evolution, we examined the correlations between averaged PDSI for the up- and downstream regions, respectively, and various rainfall-related variables (Fig. 10). The upstream drying trend correlated mostly with rainfall intensity, maximum number of consecutive wet days, amounts of light rain, and the number of days without rain (all with |r| >0.5). Conversely, the downstream wetting trend was due primarily to changes in rainfall intensity

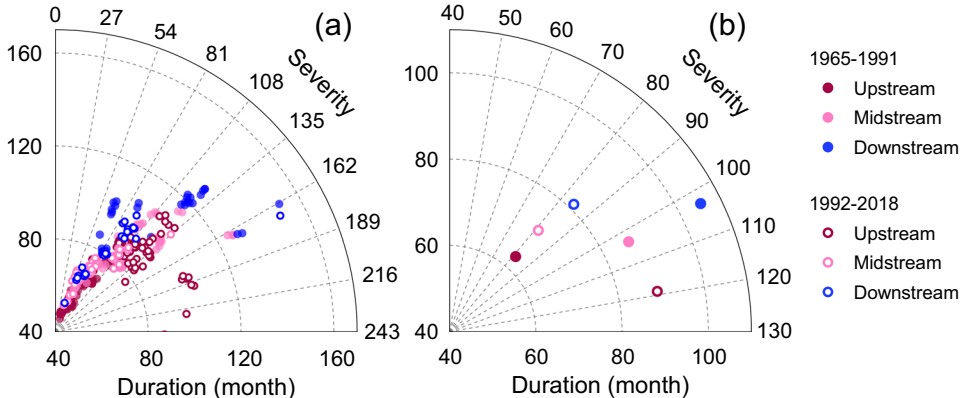

**Fig. 9 | Distribution of drought duration and severity at the subbasin and sub-region (up-, mid-, and downstream) levels in the West River Basin during the first (1965–1991) and second (1992–2018) half of the study period. a** Distribution of drought duration and severity at the subbasin level. There are 187 data points in the panel representing the subbasins we delineated in the basin. **b** Distribution of drought duration and severity at the sub-region level. Source data are provided as a Source Data file.

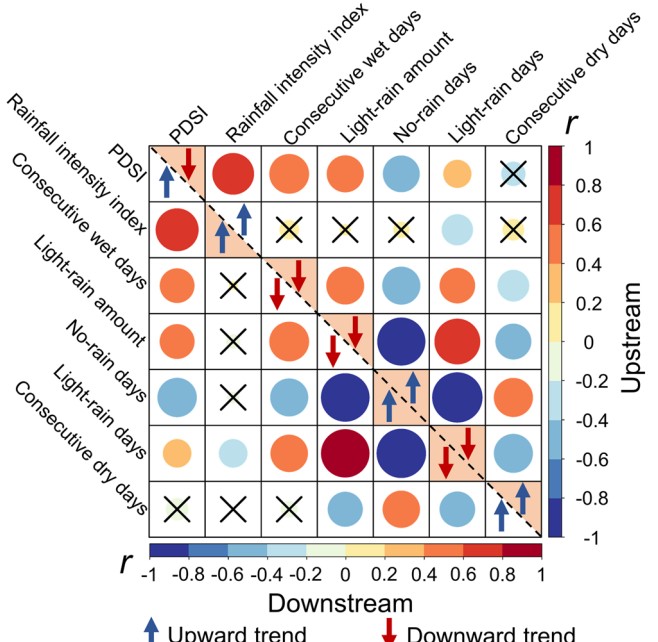

**Fig. 10 | Correlation relationship between Palmer Drought Severity Index (PDSI) and various rainfall-related variables in the upstream (upper right-hand panel) and downstream (lower left-hand panel) parts of the West River Basin.** Blue upward arrows refer to an upward trend for a specific variable, while the opposite is indicated by red downward arrows. Filled triangles indicate a statistically significant trend. The color and size of a circle refer to the correlation coefficient between PDSI and a specific variable, with red and blue indicating positive and negative significant correlations, respectively, and a black cross for non-significance. Source data are provided as a Source Data file.

and the number of days without rain (Fig. 10). Under the same rainfall pattern change, a significant decrease in continuous wet days, amounts of light rain, and a significant increase in the number of days without rain could cause upstream to become drier; whereas a significant increase in rainfall intensity and the number of days without rain may lead to more severe flood risks downstream. Finally, to examine to what extent rainfall intensification alone may explain the spatially divergent hydrological responses between the upstream and downstream parts of the WRB, a series of virtual experiments were conducted across a range of scales from individual hillslopes to HRUs, subbasins including upstream versus downstream.

## Primary mechanism of upstream drying and downstream wetting

First, we examined the hydrological response at the slope scale. A case study comprising sloping land subject to two rainfall intensity scenarios (RI = 9 and 11) was designed whilst keeping all other factors as well as the total amount of rainfall the same (1,605 mm). SWAT-modeled hydrological responses (Supplementary Fig. 4) showed greater surface runoff under increased RI, both in volume (from 164 mm to 191 mm) and proportion (from 10% of rainfall to 12%), whereas infiltration was correspondingly reduced (from 90% to 88%). As such, flooding risk at this scale increased for increased RI.

Second, we examined the hydrological response at the HRU scale. Two small watersheds considered representative of topographic conditions in the up- and downstream parts of the WRB−steeper in the highland, gentler in the lowland−were selected (see the aerial views and densities of slope and Topographic Wetness Index (TWI) as shown in Supplementary Fig. 5). Hydrological simulations were carried out using the respective rainfall time series for the two watersheds during the full study period (1965–2018) as input. It is noted that there is only one weather station for this specific watershed, ensuring the climate conditions (e.g., precipitation and temperature) are the same for all the sloping lands in the watershed. Under the prevailing intensifications of rainfall, soil water trends at HRU level proved to be dependent on slope steepness and the so-called TWI, such that steep HRUs (upland) tended to become drier, and gentle HRUs (lowland) wetter (Supplementary Fig. 5) in both watersheds.

Third, we examined the hydrological response at the sub-region scale (i.e., the entire upstream and downstream). Moving up in scale, we selected two years with closely comparable rainfall totals but strongly different intensities: 1970 (upstream rainfall and RI: 825 mm and 6.2; downstream rainfall and RI: 1605 mm and 9) and 2010 (upstream rainfall and RI: 849 mm and 6.5; downstream rainfall 1650 mm and 11) because the amounts of rainfall in these two years are very close (<3% difference) but with clear intensification during 40 years. For the sake of fairness, we further removed the 3% difference in rainfall amount by decreasing this amount for 2010 rainfall timeseries (i.e., updated 2010 rainfall became 825 mm for upstream and 1605 mm for downstream; Supplementary Table 2). Keeping all input data between the two years the same except for rainfall, the simulated hydrological responses showed annual amounts of soil water, water yield, surface runoff, and baseflow for the upstream part of the WRB all to become less under increased rainfall intensities (by −0.5%, −6%, −7%, and −8%, respectively) whereas the responses increased for the downstream part (by 0.9%, 10%, 15%, and 3%, respectively; Supplementary Table 2). Further, seasonal peak streamflow at the height of

the rainy season (June–August) increased by 51% and 15% for the up- and downstream parts, respectively, whereas minimum dry-season flows (December–February) became either less (−2% for the upstream part) or increased (by 25% downstream; Supplementary Table 2). In short, these model results can provide strong support for the contention that the intensification of rainfall in the WRB is likely the chief factor underlying the observed spatially diverging trends for water availability in recent decades, mainly as a result of contrasts in the interaction between rainfall intensity, slope gradient and slope morphology (Supplementary Fig. 5).

## Discussion

The PDSI is a widely accepted measure to characterize the dynamics of hydro-meteorological droughts, as the index integrates the water budget and soil moisture status, thereby enabling the incorporation of the effects of numerous driving factors in a comprehensive manner[38,64]. Here, spatiotemporal series of the PDSI were derived using a combination of spatially distributed hydrological modeling (with SWAT) to quantify soil moisture dynamics at the scale of hydrological response units (HRUs) and computing the associated PDSI[66]. As shown in Supplementary Fig. 6, SWAT-based simulated monthly streamflow matched observations at three gaging stations (Fig. 1) very well (including most peaks and low flows), both for the calibration (1991–2000) and validation (2001–2010) periods. With generally low values for the percentage bias (|PB| < 11%) and high to very high values for the Nash-Sutcliffe Efficiency (NSE) and $R^2$ (ranging from 0.74 to 0.95; Supplementary Fig. 6), these streamflow simulations can be rated as "good" to "very good" (Methods)[67]. As a further internal check of model performance, we validated SWAT's ET simulations using a remote sensing-based re-analysis ET product during 1980–2017 with an $R^2$ of 0.87, indicating a good model performance (Supplementary Fig. 7). For soil water, comparisons between SWAT-based simulations and two sets of satellite-based products indicated satisfactory model performance ($R^2 = 0.55$–0.56, Supplementary Fig. 8). Therefore, the dynamics of the hydrological components simulated by SWAT provided a sound basis for the PDSI calculations used in the drought analysis. We also found that basin-wide average annual PDSI values compared reasonably well with other drought indices, such as the Standardized Precipitation Index (SPI), the Standardized Precipitation Evapotranspiration Index (SPEI), and the self-calibrating PDSI (Methods; Supplementary Fig. 9).

Trend analysis highlighted a clear shift in the climate of the WRB that was characterized by intensifying rainfall and warming air (Fig. 2 and Supplementary Fig. 1). Because annual rainfall totals generally did not change appreciably (Supplementary Fig. 1a), this intensification manifested itself in the form of increases in average amount of rainfall per rain day (rainfall intensity index), the number of no-rain days, and the maximum number of consecutive dry days, but also as decreases in amounts of light rain, the number of days with light rain, and the maximum number of consecutive wet days (Fig. 2). The resulting hydrological changes as evaluated using SWAT suggested an overall mild decrease in soil water storage across the basin, along with decreased year-round surface runoff and baseflow in the upstream part, and increased wet season runoff in the downstream part (Fig. 4). Thus, the upstream area became drier in terms of soil water and streamflow in both seasons, whereas the mid- and downstream regions became wetter in the wet season (with increased surface runoff). This spatial contrast could not be explained by the observed overall warming, as the latter did not show a distinct spatial pattern (Supplementary Fig. 1b, c). Rather, the increasingly intense and temporally more concentrated rainfall raised amounts of quick-response surface runoff during the wet season, whereas the reduction in amounts of light rain, together with the more elongated dry periods, reduced soil water storage and groundwater recharge, leading to reduced baseflow (i.e., reduced streamflow in the dry season), especially upstream.

Therefore, rainfall intensification in the broad sense of the term can explain the spatiotemporal pattern of the associated hydrological responses identified by the process modeling, as also demonstrated convincingly by the results of rainfall intensity scenario analyses across a range of spatial scales (Supplementary Figs. 4 and 5; Supplementary Table 2). In addition, the observed wetting of the downstream part of the WRB implies a higher risk of flooding given the generally positive correlation between flood occurrence and soil moisture status[68–74].

Our long-term time series of PDSI-values based on spatially distributed hydrological process modeling not only gave a general picture of drought evolution across the WRB with time, but also allowed derivation of the drought patterns at an unprecedented level of spatial detail and flexibility (from HRUs to subbasins and regions; Fig. 5). The results are generally in line with previous reports of the drought situation in Southwest China[52,75] which emphasized the shortage of water for domestic and agricultural uses and the substantial socio-economic losses associated with drought[50]. Further, the EOF analysis not only illustrated the divergent spatial pattern of drought evolution across the WRB with drying upstream and wetting downstream, but also detected a shift in drought regime around 1992. For the upstream region, there was a significant positive trend in the extent of both mild and moderate droughts (Supplementary Fig. 2a, b) as well as a clear shift towards increased drought frequencies (of all grades) after 1992 (Fig. 7). In addition, both drought duration and severity increased in the upstream part, while decreasing in the mid-stream and, particularly, the downstream parts (Figs 8 and 9).

Similar changes in rainfall patterns as the ones observed here (e.g., decreases in light rain amount and days, increases in rainfall intensity and no rain days) have been reported previously for some areas within the WRB[7,76] and elsewhere in the globe[77–79]. Shifts in seasonal precipitation cycles can lead to substantial changes in the frequencies of droughts and floods[80], even in the absence of any trends in annual totals. Increases in heavy rainfall typically increase surface runoff and may lead to severe flooding, whereas decreases in light and moderate rainfall can prolong dry periods and increase drought risk[78]. Although the phenomenon of upstream drying and downstream wetting identified for the WRB has also been reported for other river basins around the world, such studies—unlike the present work—limited to descriptions of the changes in streamflow and/or total water storage[81–83], with no further attempt at identifying the primary underlying mechanism (cf. Supplementary Fig. 5). In brief, our study emphasized the causality between spatially divergent hydrological changes and rainfall intensification, highlighting the exacerbated drought and flooding and their interrelation in a river basin.

Being located in the southern part of China, the WRB has a seasonal monsoon climate with maximum precipitation in summer and a dry winter/spring season. As such, summer flooding and drought conditions during spring are a regular phenomenon in the region[49,84]. However, the results of the present study demonstrated that intensifying rainfall has led to important changes in the spatiotemporal distribution of hydrological responses, notably significant drying of the upstream parts of the basin versus wetting of the downstream parts (Figs. 4, 5, and 9). These hydrological changes cause a series of problems. Firstly, they exacerbate the problems inherent to the monsoonal climate—the dryness in dry season and wetness in wet season. Secondly, the upstream drying and associated reduction in water availability (baseflow: Fig. 4g, h; minimum flows: Supplementary Table 2) will increase the discrepancy between water demands and supply, thereby potentially causing greater agricultural and socio-economic losses. Thirdly, the increases in surface runoff and maximum streamflow in summer, together with the increased soil wetness in some areas (Fig. 4; Supplementary Table 2) will raise the flooding risk in terms of degree and areal extent in the downstream part of the basin. Consequently, the identified spatial pattern of upstream drying upstream and downstream wetting exacerbates the existing water-

related problems in the basin and heightens socio-economic pressures on regional water resource managers.

This study investigated the hydrological responses to changes in rainfall pattern in recent decades across the West River Basin, a large (353,120 km²) watershed in monsoonal Southern China. The intensifying rainfall over the 54-year study period (1965–2018) was reflected by clear and significant increases in amount of rainfall per rainy day, the number of days without rain, and the maximum number of consecutive dry days, but also by concurrent decreases in amount of light rain, the number of days with light rain, and the maximum number of consecutive wet days. Combining this climatic information with spatially distributed hydrological process modeling yielded a clear spatial pattern of changes in dryness and wetness across the basin. The inferred decreases in annual water yield and surface runoff for the upstream region, and increases thereof for the downstream region, indicated increasingly severe water shortages upstream, as well as greater overall water availability and higher flood risk downstream in recent decades. Further, combining the hydrological process modeling with computation of the Palmer Drought Severity Index allowed derivation of patterns of drought occurrence, extent, and severity at an unprecedented level of spatial detail (down to 2738 so-called Hydrological Response Units), thereby supporting improved representation at larger scales (i.e., subbasin and region). Both analysis of the spatiotemporal evaluation of drought and Empirical Orthogonal Function analysis, confirmed the severe drying trend for the upstream part of the basin versus the wetting of the downstream part inferred by the hydrological modeling. Finally, similarly diverging trends in dryness/wetness and water availability for the up- and downstream areas were confirmed in a virtual experiment that isolated the hydrological effects of rainfall intensification across various scales. Although the present study focused on a single large river basin in Southern China, our finding that a temporal change of rainfall (i.e., intensification of rainfall) can cause major spatially contrasting changes in hydrological response due to differences in slope gradients and curvature may also apply to other large river basins experiencing comparable levels of climatic intensification, with major implications for the exacerbation of both drought and flooding risks.

## Methods

### Study area

The West River Basin (WRB) is a macro-scale watershed (353,120 km²) located in South China, accounting for 78% of the Pearl River Basin in area with a significant East Asian monsoon climate (Fig. 1)[85,86]. With a total length of 2214 km, the main stem of the West River originates in the Maxiong Mountains in Yunnan province and runs generally from West to East through the provinces of Guizhou, Guangxi, and Guangdong before entering the South China Sea in the Pearl River Delta (Fig. 1)[84]. The terrain is characterized as mountainous with intramontane plateaux in the northwest, consisting of small to medium mountains and hills in the central part, and becoming increasingly lower and flatter in the downstream part[87]. The bedrock geology consists primarily of granites, sandstones, shales, and limestones of Precambrian and Paleozoic age, respectively, whereas Quaternary alluvial deposits are important as well, particularly in the downstream region. The dominant soil types include latosolic (37.7%), calcareous (12.2%), and alluvial soils (15.3%)[88,89] which broadly correspond to the ferralsols/acrisols, rendzinas, and fluvisols, respectively, in the UNESCO/FAO classification. The primary land-cover types are evergreen forests−both natural monsoon broad-leaf forest, regenerating pine forests and coniferous plantations (61%)−grasslands (15%), and farmland (22%). Depending on elevation, the WRB has a sub-tropical to tropical monsoon climate, with average annual temperatures ranging from ~14 °C in the western headwater area to ~22 °C in the eastern lowlands[85]. Mean annual precipitation varies from ~1000 mm upstream to ~2200 mm downstream, with an overall mean of ~1480 mm. The

average annual streamflow at the Tiane gaging station on the upper main stem is 357 mm (mean discharge of 1545 m³ s⁻¹), compared to ~640 mm at Gaoyao close to the basin outlet (Fig. 1). About 80% of streamflow is recorded during the main monsoon season (April to September)[85,90]. The West River is the second-largest in China in terms of streamflow amount, with an average discharge about five times that of the Yellow River, or 4.5 times that of the European Rhine River[91].

### Observational climate and streamflow data

Quality-controlled daily meteorological data for 31 stations covering 54 years (from 1965 through 2018) were obtained from the Data Center of the China Meteorological Administration (http://data.cma.cn), including rainfall, maximum and minimum air temperatures, relative humidity, and wind speed. Solar radiation inputs were estimated from measured sunshine duration data as follows:

$$Q = Q_A\left(a + b\frac{n}{N}\right) \tag{1}$$

where $Q_A$ is the maximum possible radiation at the top of the atmosphere, $n$ is sunshine duration, $N$ is the maximum possible duration of bright sunshine (both in hours), and $a$ and $b$ are empirical constants.

Streamflow data is measured daily by an agency affiliated with the Pearl River Water Resources Commission; we used the quality-controlled monthly data for three streamflow-gaging stations (Tiane, Wuxuan, and Gaoyao; see Fig. 1 for locations) for hydrological model calibration and validation (see the corresponding section below).

### Rainfall intensification metric

To represent rainfall intensity, we defined and examined six metrics derived from the daily time series of rainfall (RF), i.e. rainfall intensity (RI, average rainfall amount per rain day); light-rain amount (rainfall accumulation for RF < 10 mm d⁻¹); light-rain days (number of days with RF < 10 mm d⁻¹); no-rain days (number of days without rainfall, used as an indirect measure of rainfall frequency in a year); the maximum number of consecutive dry days; and the maximum number of consecutive wet days in a year. The first two metrics (RI and light-rain amount) represent the intensity of rainfall at the daily scale, while the remaining metrics of rainfall frequency can be regarded as characterizing the intensity of rainfall at an annual scale.

### Spatially distributed hydrological modeling: SWAT model

The Soil and Water Assessment Tool (SWAT) model was developed by the U.S. Department of Agriculture's Agricultural Research Service (USDA–ARS) for the investigation of the effects of climate and land management practices on water, sediment, and agricultural chemical yields[92–94]. This process-based watershed-scale model simulates the water budget, plant growth, transportation of sediment and agricultural chemical yields at a daily time step. The hydrological part of the model is based on the water balance of the soil profile and includes precipitation, surface runoff and infiltration, evapotranspiration (ET), soil water movement (lateral flow and vertical percolation), and baseflow[95]. Here, the Penman-Monteith method was selected for the estimation of potential evapotranspiration (PET), whereas the model uses a daily leaf area index to partition the PET into potential soil evaporation and potential plant transpiration[95]. Further details can be found in the theoretical and practical descriptions of the model[93,96].

### SWAT: model input, setup, and calibration/validation

SWAT requires input data for the characterization of climate, topography, soil, land cover, and land management[97]. ArcSWAT (version 2012) was employed to generate the respective input files and various watershed property-related parameters (e.g., soil texture, bulk density, hydraulic conductivity, available water capacity). In a SWAT setup, a river basin (i.e., a study area) is delineated to a number of subbasins

based on topography, and a subbasin is divided into multiple hydrological response units (HRUs) based on unique combinations of land use, soil type, and slope in the subbasin[92,93]. SWAT adopts an equation to adjust the Soil Conservation Service (SCS) Curve Number (CN) to a different slope, reflecting increased overland flow for a higher slope[98]. For the WRB, Shuttle Radar Topography Mission (SRTM) digital elevation model (DEM) data at 90-m resolution[99] were used to delineate the WRB. The land-cover changes in the WRB between 1980 and 2015 were quite minor, hence the SWAT setup used the land-cover map for year 2000 (1 km × 1 km resolution) to represent the full study period. The soil map was obtained from the Ecological and Environmental Science Data Center for West China. In this study, the WRB was delineated into 187 subbaisns and 2738 HRUs. The least computation unit of the model is HRU, then simulated hydrological components (e.g., water yield, ET, soil water, surface runoff, baseflow) of all HRUs within a subbasin were integrated to give the corresponding results per subbasin. A similar integrative approach was used for the larger scales (i.e., the upstream, mid- and downstream, and the entire WRB).

SWAT-CUP (Calibration and Uncertainty Programs) SUFI-2[100–102] was used to calibrate the model's key parameters using streamflow data for three gaging stations—Tiane, Wuxuan, and Gaoyao that represented the upper, middle, and downstream reaches, respectively (Fig. 1 and Supplementary Fig. 6). For each station, 10-year streamflow records were used for calibration (1991–2000) and subsequent ten-year records for validation (2001–2010; with 1 year (2010) of observed streamflow missing at Tiane (Supplementary Fig. 6)). A 5-year warming-up period (1960–1964) was applied to minimize the impacts of uncertain initial conditions on model simulations for the full 1965–2018 period. Based on our previous work with SWAT calibration[102–105], we selected five key parameters for calibration for the up-, mid-, and downstream parts, respectively, and the resulting fitted parameter values are listed in Supplementary Table 3. In addition, we did internal checking of model simulations of ET and soil water, using re-analysis ET data based on a fusion of ERA5, MERRA2, and GLDAX2-Noah data during 1980–2017[106] and satellite-based root zone soil water amount from Centre Aval de Traitement des Données SMOS (CATDS) during 2011–2018[107] and GLEAM-based topsoil water during 1980–2018[108,109], respectively.

## Drought assessment

Various widely-used statistical drought indices (e.g., the Standardized Precipitation Index (SPI)[110], the Standardized Precipitation Evapotranspiration Index (SPEI)[111], and the Streamflow Drought Index (SDI)[112]) use only one or two climatic variables. In contrast, the Palmer Drought Severity Index (PDSI)[113] requires more input data (e.g., ET, soil available water capacity, soil water content, and water yield) to drive a two-layer water balance model and evaluate the degree of soil moisture stress on a monthly time scale. The use of the PDSI facilitates comparisons across time and space and is capable of capturing extreme events because values are normalized by average moisture conditions[31,38,64,114]. Hence, the PDSI was selected to assess drought changes across the WRB.

We developed a watershed-scale drought evaluation system that uses SWAT-based spatially distributed hydrological modeling to compute hydrological variables that are required for PDSI[115] calculation. In this system, the least computation unit is HRU (there are 2738 HRUs in the WRB). For each HRU, the spatially-distributed SWAT modeling can provide dynamics of soil water content, PET and ET, water yield, and recharge. Then, these hydrological variables were taken as input to compute PDSI for each HRU at monthly time scale. Finally, the HRU-based results can be summarized into sub-basin, basin, and even regional scales. In addition, to evaluate the performance of PDSI, the basin-wide annual PDSI values were compared with SPI, SPEI, and scPDSI (obtained from the Climatic

Research Unit (CRU))[116]. The detailed procedure of the SWAT-based drought evaluation system and PDSI-calculation equations can be found in our previous publication[66].

We used a universally-adopted PDSI-based drought classification system that can help identify occurrence of a drought event and recognize its grade as mild, moderate, severe, or extreme (Supplementary Table 4). Drought frequency is defined as the number of drought events during a given period of time[117] or is expressed as the ratio of the time with drought occurrence to total study time. The so-called 'runs' theory was developed to characterize various drought features, including duration, intensity, and severity[118]. A run is defined as a time series of variables indicating drought (e.g., PDSI). If the values in the run are all below a given threshold (e.g., PDSI= −1), the run is considered negative, otherwise the run is positive[5,31]. Drought duration refers to a period in terms of weeks, months, years or other time segments during which all drought indices are below a given threshold[119]. Drought severity refers to the sum of the portions of drought indices that are lower than the given threshold throughout the duration. Here, we used the runs theory and the spatiotemporal series of PDSI to evaluate drought duration and severity across the WRB.

The Empirical Orthogonal Function (EOF) method is a statistical decomposition approach that extracts useful information by reducing dimensionality[120–123]. It is widely used for the identification of dominant spatiotemporal patterns of geophysical variables, especially in the field of global change[121,124–126]. A set of EOF for a specific variable or data-set with $m$ observations at $n$ stations (giving an $m \times n$ matrix X), can be denoted as:

$$X(t,s) = \sum_{i=1}^{n} PC(s) \times EOF(t) \qquad (2)$$

where $X(t,s)$ is the original spatiotemporal series or dataset as a function of time (principal components, PCs) and space (EOF modes, EOFs), and $n$ is the sample size of space.

In this study, we used the EOF method to decompose spatiotemporal series of drought by analyzing the first two EOF modes (spatial pattern) and the related temporal variability (PCs). For this, we adopted the annual series of PDSI values for the 187 subbasins of the WRB over the 54 years of observations, giving a matrix size of 54×187. The North test was used to identify significant EOFs[65]. An EOF was considered significant if its error range did not overlap with that of the next higher EOF.

## Trend analysis

We adopted the widely used Mann-Kendall test to analyze the results decomposed by EOF[127,128]. We also used the Pearson chi-square normality test to check the distribution of key climate variables and drought indices[129], after which linear regression analysis was employed in combination with the least-squares method to detect any trends. If the slope of a fitted linear line differed significantly from zero (t-test: $P < 0.05$), the trend was regarded as statistically significant. Trends for the respective climate elements over 1965–2018 study period were analyzed per climate station; the inverse distance weighted interpolation method[130] was used to present the spatial distributions. Time series of monthly PDSI values at subbasin/HRU level were used to detect annual and seasonal trends of drought.

## Data availability

For SWAT input and setup, daily meteorological data for 31 stations are obtained from the Data Center of the China Meteorological Administration (http://data.cma.cn). The DEM data is downloaded from the Shuttle Radar Topography Mission (https://srtm.csi.cgiar.org). The land use data are collected from the Resources and Environmental Science Data Center (https://www.ncdc.ac.cn). The soil map is

obtained from the Ecological and Environmental Science Data Center for West China (http://westdc.westgis.ac.cn). For calibration and validation of SWAT, daily streamflow data is obtained from an agency affiliated with the Pearl River Water Resources Commission. The re-analysis ET data are available at the National Tibetan Plateau Data Center (https://www.tpdc.ac.cn). The SMOS L4 RZSM product is obtained from the Centre Aval de Traitement des Données SMOS (https://www.catds.fr). The GLEAM soil moisture data are available at the Global Land Evaporation Amsterdam Model (https://www.gleam.eu). For the evaluation of PDSI performance, the scPDSI data is obtained from the Climatic Research Unit (https://crudata.uea.ac.uk/cru/data/drought/). Source data are provided with this paper.

## Code availability
The code used in this study is available from the authors upon request.

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

## Acknowledgements

Y.P.W. acknowledges support from the Strategic Priority Research Program of the Chinese Academy of Sciences (Grant No. XDB40020205) and the National Science Foundation of China (Grant No. 42271025 and Grant No.31961143011). G.Y.Z. acknowledges support from the National Science Foundation of China (Grant No. 42071031 and Grant No. 42130506). Y.P.W. acknowledges support from the Innovation Team of Shaanxi Province (Grant No.2021TD-52) and the Shaanxi Key Research and Development Program (Grant No. 2022ZDLSF06-04). We also thank Ms. Xiuwen Chen and Mr. Shantao An for their participation into modeling work.

## Author contributions

Conceptualization: G.Y.Z., Y.P.W. Methodology: Y.P.W., G.Y.Z., X.W.Y., F.W. Investigation: X.W.Y., Y.P.W. Visualization: X.W.Y., Y.N.S., F.W., G.C.Z. Supervision: G.Y.Z., Y.P.W. Writing—original draft: Y.P.W., X.W.Y., G.Y.Z., L.A.B. Writing—review and editing: G.Y.Z., Y.P.W., L.A.B., A.G.D., P.G., X.W.Y., F.W., and D.C.Z.

## Competing interests

The authors declare no competing interests.
