## [Peer Review File · Nature Communications]

Rising rainfall intensity induces spatially divergent hydrological changes within a large river basinReviewers' comments:

Reviewer #1 (Remarks to the Author):

The manuscript entitled "Rising rainfall intensity induces simultaneous upstream drought and downstream flood of a basin" by Wu et al investigates the features and mechanism of spatiotemporal changes in dryness and wetness in the West River Basin, South China. First, the authors analyzed annual trends in key climatic variables. Second, a hydrological model (SWAT) was used to investigate changes in hydrological components. Third, the drought index PDSI at monthly timescale was computed for sub-basin and HRU levels. Finally, EOF analysis was applied on the PDSI to investigate the spatiotemporal dynamics of drying and wetting. Overall, the authors claim that increases in rainfall intensity led to significant drying in the upstream (causing drought) but wetting in the downstream (causing flood) parts of the study basin.

The topic of the study is interesting. The methods seem appropriate to address the research problem as well. However, I have a few concerns that need to be addressed before considering for publication.

1. The results of this study indicated that downstream is getting wetter (i.e., increasing surface runoff, soil moisture) at annual time scale. This may increase flood risk, but it is not relevant directly to actual floods that usually happen at event-scales. Further discussion is needed to clarify this point and make the conclusion of the study more robust.

2. The authors need to do a better job for selecting appropriate colormaps for each figure. It is difficult for reader to interpret some results because of the poor visualization. In many figures, try to have the same color scale if possible for better comparison between sub-plots. For instance, in Figure 4 the same color scale should be used for subbasin and HRU for each variable. For colormap that shows both negative and positive trends (same magnitude), keep the colormap symmetrical with white color in the middle (around 0).

Other comments:

- a. Figure 1: What is the blue line in the inset figure at the lower right corner? It is outside the study river basin (which is the red polygon). Also, why do the authors include the nine-dash line map in South China Sea here (bottom right corner of the inset)? This information is irrelevant to the study.
- b. Figure 5: Why the unit of annual trend is $001/y$? Should it be y^{-1} ?
- c. Figure 8: The labels for sub-basin and HRU should be added. Remember to use the same color scale.
- d. Figure 9: Do color dots in panel (a) represent 187 sub-basins? This should be mentioned clearly in the caption.

Reviewer #2 (Remarks to the Author):

The manuscript is interesting and presenting the rainfall changes impacts on the hydrological components, combining drought indexes and the ArcSWAT hydrological model. However, some modifications are needed.

In general, the English language should be improved. Furthermore, series of recommendations are suggested below:

Title

- Please modify the title

Abstract

- The abstract should be more detailed. Please, give a more numerical description of the results.

Introduction

- Please outline the innovation of this work

Results: the results are well written.

- Figure S1: please use different colour scale ramp for the parameters

Conclusion

- Line 411: his study investigated the hydrological responses to climate change: In my opinion this indicates the use of climate model data.... Please rephrase...for instance changes in rainfall pattern

Discussion

- Please add a paragraph comparing your results to other similar papers results.

Material and Methods

- More detail description about the study area. A presentation about the morphology and land use should be added.
- Line 460: climate data instead of climatic data
- Line 522-524: please rephrase
- Line 527-531: the sentence should be rewritten
- Line 544-546: Please briefly describe it
- Line 572-575: Please rewrite
- Line 579: please rewrite

Reviewer #3 (Remarks to the Author):

This manuscript investigated changes in dryness and wetness between 1965 and 2018 over the West River Basin in South China, mostly based on simulation results from a calibrated regional hydrological model (SWAT). The authors suggest that rainfall intensification across the basin can cause simultaneous increase of droughts and floods in different parts of the basin. Although the message has important implications for mitigation and adaptation under climate change, my main concern with this study is that the causality of rainfall intensification and increasing droughts and floods is not well supported by sufficient evidence and reasoning. If rainfall intensity increase is similar both upstream and downstream,

why is the hydrological response divergent? Under what condition would other basins also expect simultaneous increase of floods and droughts? While the paper is generally well written as a case study, without much improved in-depth analyses of hydrological balance and related mechanisms that lead to the observed changes, I cannot recommend publication of the manuscript in Nature Communications.

I am also not fully convinced with the robustness of the results, as they are based on simulations from only one model. Although the model is calibrated against streamflow from three gauges reasonably well (one might however question the model performance at Tiane, where the simulated minimum flow is notably lower than observation in validation period), no comparison is made to evaluate if the model can also reproduce other parts of the hydrological cycles (evapotranspiration, soil moisture, water balance etc.) reasonably well. There are many remote-sensing based products for such evaluation.

Additionally, parts of the methodology and results interpretations need better explanation:

1. What is the link between HRU and subbasin in the simulations, or how is simulated results from HRUs upscaled to subbasins?
2. What is the optimization criteria in calibration, and are the calibrated parameters fixed for the whole region?
3. Would the EOF analysis leave out important patterns when the cumulative explained variance is below 51% for the first two EOFs?
4. Are differences in hydrological response between 1970 and 2010 due to rainfall intensification alone? Are there any dams or other human driven changes between 1970 and 2010 in this basin?
5. Why it is concluded that wetting would occur downstream (instead of over the full basin), when the change in discharge is much more drastic (>13 times) upstream? Or are the two years not representative of the changing rainfall intensity in this basin?

I would leave out the minor issues so far, except for pointing out that in Table S2, data in the last row is incorrect.

Response to Reviewer's Comments

Reviewer 1

The manuscript entitled “Rising rainfall intensity induces simultaneous upstream drought and downstream flood of a basin” by Wu et al investigates the features and mechanism of spatiotemporal changes in dryness and wetness in the West River Basin, South China. First, the authors analyzed annual trends in key climatic variables. Second, a hydrological model (SWAT) was used to investigate changes in hydrological components. Third, the drought index PDSI at monthly timescale was computed for sub-basin and HRU levels. Finally, EOF analysis was applied on the PDSI to investigate the spatiotemporal dynamics of drying and wetting. Overall, the authors claim that increases in rainfall intensity led to significant drying in the upstream (causing drought) but wetting in the downstream (causing flood) parts of the study basin. The topic of the study is interesting. The methods seem appropriate to address the research problem as well. However, I have a few concerns that need to be addressed before considering for publication.

[Response] We thank the Reviewer for his/her very positive evaluation of our work, and we respond to the concerns in details as shown below.

1. The results of this study indicated that downstream is getting wetter (i.e., increasing surface runoff, soil moisture) at annual time scale. This may increase flood risk, but it is not relevant directly to actual floods that usually happen at event-scales. Further discussion is needed to clarify this point and make the conclusion of the study more robust.

[Response] We thank for the good comment, and we agree ‘getting wetter’ is not relevant directly to ‘actual flood’. Following the reviewer’s suggestion, therefore, we have added further discussion to clarify this point—getting wetter (e.g., increasing surface runoff, soil moisture, more baseflow) may increase the flood **risk** instead of actual flood. Please see a brief statement on P9 L402-405 and detailed

discussions as shown below.

**(1) Soil moisture condition plays a vital role in driving floods**

Soil moisture state was identified as a primary driver in flood generating
mechanisms for watersheds in Europe (Berghuijs et al., 2019), the United States
(Berghuijs et al., 2016), Bangladesh (Han et al., 2021), and other parts of the
world (Sharma et al., 2018; Wasko et al., 2020). In contrast, the decrease in soil
moisture led to a decline in observed flood discharge even with increased
extreme precipitation in Australia (Wasko and Nathan, 2019) and North China
(Yang et al., 2021). Thus, there may be a positive correlation between floods
and soil moisture.

**(2) The correlation between floods and water yield (streamflow)**

Long-term trend analyses found that the flooding trend is aligned with that of
streamflow and soil moisture (Wasko et al., 2021), and flash flood susceptibility
could directly increase with rising surface runoff (Mahmood and Rahman, 2019),
which is the primary contributor to flood owing to its property of quick-response.
Further, slow-response baseflow can shape flooding because a wetter landscape
tend to disproportionally discharge rainfall to rivers, leading to a larger flow rate
(Kendall et al., 1999), and a long-term trend in flood magnitude tends to align
with that of groundwater storage and baseflow (Berghuijs and Slater, 2023).
Therefore, there might be a positive correlation between floods and water yield,
including the overland surface runoff and baseflow.

In brief, the above reported positive correlations could tell a higher flood **risk** under
a higher soil moisture condition and increased water yield.

2. The authors need to do a better job for selecting appropriate colormaps for each
figure. It is difficult for reader to interpret some results because of the poor
visualization. In many figures, try to have the same color scale if possible for better
comparison between sub-plots. For instance, in Figure 4 the same color scale should
be used for subbasin and HRU for each variable. For colormap that shows both
negative and positive trends (same magnitude), keep the colormap symmetrical with

white color in the middle (around 0).

**[Response]** We thank for the good comment. Following the Reviewer's suggestion,
we have updated Fig. 4 (as shown below), Fig. 5, and Fig. 8 (please see our
**Responses to Questions b and c of Other Comments**) through using the same color
scale.

**Figure 4.** Spatial distribution of annual trends for four hydrological components
across the West River Basin.

**Other comments:**

a) Figure 1: What is the blue line in the inset figure at the lower right corner? It is
outside the study river basin (which is the red polygon). Also, why do the authors
include the nine-dash line map in South China Sea here (bottom right corner of the
inset)? This information is irrelevant to the study.

**[Response]** The blue lines in the bottom right large inset figure are two major
rivers—the Yangtze River and the Yellow River. To avoid confusion, we removed
these two lines in the modified figure. The purpose of the two inset figures at the
bottom right corner was to display the geographical location of the study area (the

West River Basin) in China, and now we have reduced the size and removed one
 inset figure (please see new Fig. 1 as shown below).

 Figure 1. Location of the West River Basin (WRB) within South China plus locations
 of climatic and stream gauging stations used in the present analysis.

b) Figure 5: Why the unit of annual trend is 001/y? Should it be y^{\wedge} ?

**[Response]** Thanks for the good catch. The magnitude of change rate of PDSI is
 quite small, then using a small unit ($0.01\ y^{-1}$) can help readers get variation of
 drought index. Following the Reviewer’s suggestion, we have used y^{-1} in the revision
 of the figure as shown below.

 Figure 5. Spatial distribution of annual trends of the PDSI at subbasin and
 hydrological response unit (HRU) level in the West River Basin.

c) Figure 8: The labels for sub-basin and HRU should be added. Remember to use the
 same color scale.

**[Response]** We have modified Fig. 8 following the Reviewer’s suggestion.

Figure 8. Spatial distribution of drought duration and severity at the subbasin and
 hydrological Response Unit (HRU) levels in the West River Basin.

97 d) Figure 9: Do color dots in panel (a) represent 187 sub-basins? This should be
 mentioned clearly in the caption.

**[Response]** Yes, the color dots represent 187 subbasins, and we have clarified this
 point in the figure caption.

**Reviewer 2**

The manuscript is interesting and presenting the rainfall changes impacts on the
 hydrological components, combing drought indexes and the ArcSWAT hydrological
 model. However, some modifications are needed. In general, the English language
 should be improved. Furthermore, series of recommendations are suggested below:

**[Response]** We thank the Reviewer for his/her positive evaluation of our work, and the
 writing has been improved by a native English speaker. Our responses to other review
 comments are shown below.

1. Title: Please modify the title.

**[Response]** We have rephrased the title to reflect the main finding of the study (see
 below).

Rising rainfall intensity induces spatially divergent hydrological changes within a large
 river basin

2. Abstract: The abstract should be more detailed. Please, give a more numerical
description of the results.

**[Response]** We thank for the good suggestion. Now we have presented some
important results / conclusions with some numerical descriptions (please see P1 L30-
40).

3. Introduction: Please outline the innovation of this work.

**[Response]** We thank for the good comment. In the revision, we further enhanced
the description of research gap and motivation at the end of the Penultimate
Paragraph of Introduction (please see P3 L114-120 or below).

However, it is still unclear whether, and to what extent, the drying in the Southwest and
the flooding in the Southeast are related. In particular, it is not well understood whether
the two phenomena can be attributed to a single (e.g., altering precipitation patterns) or
multiple climate factors. These questions motivated us to examine how precipitation and
other hydroclimatic fields have changed in the past decades, what the resulting
hydrological responses have been, and to what extent regional drying and flooding are
potentially interrelated.

Then we outlined the objectives and hypothesis to highlight the innovation of our
work (please see P3 L124-129 or important points as below).

(1) assess the spatiotemporal patterns of changes in precipitation and other
hydrometeorological variables

(2) quantify the associated changes in hydrological responses

(3) investigate the primary evolutionary mechanism of dry and wet spells (drought/flood
risks) across the basin and their possible interrelation

(4) test the hypothesis that rainfall intensification constitutes the primary driver of both
drought occurrence (in the Southwest) and flooding (in the Southeast)

4. Results: the results are well written.

Figure S1: please use different color scale ramp for the parameters.

**[Response]** We thank for the positive evaluation of our Results. Following the good
suggestion, we used different color scale ramps for different parameters, and the
same ramp was used for maximum and minimum Temperature (TX and TN). Please

see new Figure S1 as shown below.

Figure S1. Spatial distribution of annual trends of rainfall amount (RA; mm),
maximum, and minimum air temperature (TX, and TN; °C), wind speed
(WS; m s⁻¹), relative humidity (RH), and solar radiation (SR; MJ m² d⁻¹)
in the WRB for the 54-year (1965–2018) period.

5. Conclusion: Line 411: this study investigated the hydrological responses to climate
change: In my opinion, this indicates the use of climate model data.... Please
rephrase...for instance changes in rainfall pattern.

[Response] We thank for the good comment. Done as suggested (please see P10
L458).

6. Discussion: Please add a paragraph comparing your results to other similar papers
results.

[Response] We thank much for the good suggestion. Please see the new paragraph
on P9 L423-436 or below.

[revised manuscript text omitted]

**Reviewer 3**

**A.** This manuscript investigated changes in dryness and wetness between 1965 and
2018 over the West River Basin in South China, mostly based on simulation results
from a calibrated regional hydrological model (SWAT). The authors suggest that
rainfall intensification across the basin can cause simultaneous increase of droughts and
floods in different parts of the basin. Although the message has important implications
for mitigation and adaptation under climate change, my main concern with this study
is that the causality of rainfall intensification and increasing droughts and floods is not
well supported by sufficient evidence and reasoning. If rainfall intensity increase is
similar both upstream and downstream, why is the hydrological response divergent?
Under what condition would other basins also expect simultaneous increase of floods

and droughts? While the paper is generally well written as a case study, without much
improved in-depth analyses of hydrological balance and related mechanisms that lead
to the observed changes, I cannot recommend publication of the manuscript in Nature
Communications.

**[Response]** We appreciate this critical comment that helps us realize that we DO need
to improve the presentation substantially for convincing the reviewer and audience
about the mechanism we stated. For this purpose, we have done substantial work to
clarify the causality between drought change and rainfall intensity with evidence in this
revision, and please see below for step-by-step reasoning (please see P7-8 L316-353 or
below).

**Step 1: Increased rainfall intensity would increase surface runoff and decrease**
**infiltration for a general slope land (Increased Overland flow and Flooding Risk)**

Sloping land. A case study comprising sloping land subject to two rainfall intensity
scenarios (RI = 9 and 11) was designed whilst keeping all other factors as well as the
total amount of rainfall the same (1,605 mm). SWAT-modeled hydrological responses
(Fig. S4) showed greater surface runoff under increased RI, both in volume (from 164
247 mm to 191 mm) and proportion (from 10% of rainfall to 12%), whereas infiltration was
248 correspondingly reduced (from 90% to 88%). As such, flooding risk at this scale
increased for increased RI. This can be a universal hydrological response to increased
rainfall intensity for a sloping land.

**Figure S4. Schematic of SWAT-simulated hydrological responses to rainfall (XXX**
**station) with same amount (1605 mm) but different intensities (9 for year**
**1970 and 11 for year 2010) for a typical slope land (slope = 0.5).**

**Step 2: A higher slope may be drying and a lower slope may be wetting at HRU**
**level for both upstream and downstream (Drying Highland and Wetting Lowland)**

HRU scale. Two small watersheds considered representative of topographic
conditions in the up- and downstream parts of the WRB—steeper in the highland,
gentler in the lowland—were selected (see the aerial views and densities of slope and
Topographic Wetness Index (TWI) as shown in Fig. S5). The average slope is 18 and 7
for the upstream and downstream small watersheds, respectively; similarly, the average
TWI is 6 and 9, respectively. The topography comparison shows that the upstream
watershed is generally steeper (i.e., more highlands and less lowlands) than the
downstream one (i.e., more lowlands and less highlands). Hydrological simulations
were carried out using the respective rainfall time series for the two watersheds during
the full study period (1965–2018) as input. Under the prevailing intensifications of
rainfall, soil water trends at HRU level proved to be dependent on slope steepness and
the so-called TWI, such that steep HRUs (upland) tended to become drier, and gentle
HRUs (lowland) wetter (Fig. S5) in both watersheds. For example, the soil water trend
would become negative (decreasing soil water) if the slope is over 25 degree (or $TWI < 6$)
in the upstream watershed. That is to say, the highland (steep HRUs) would become
drying, and the lowland (gentle HRUs) would become wetting for both upstream and
downstream watersheds. This can be a universal hydrological response—divergent soil
water trends in highland and lowland—to the intensified rainfall at the HRU level.

Figure S5. Relationship between soil water trend and slope at the subbasin level for a typical small watershed in the upstream (left panels) and downstream (right panels) areas, respectively. (a and a') Density of slope distribution, (b and b') Aerial view of the watershed, (c and c') Spatial pattern of slope, (d and d') Spatial pattern of Topographic Wetness Index (TWI), (e and f, e' and f') Relationship between soil water trend (1965–2018) and slope and TWI.

Step 3: Upstream would generally become drying and downstream would be wetting owing to the intensified rainfall (Spatially Divergent Changes)

Subbasin scale (upstream/downstream). Moving up in scale, we selected two years with closely comparable rainfall totals but strongly different intensities: 1970 (upstream rainfall and RI: 825 mm and 6.2; downstream rainfall and RI: 1,605 mm and 9) and 2010 (upstream rainfall and RI: 849 mm and 6.5; downstream rainfall 1,650 mm and 11) because the amounts of rainfall in these two years are very close (<3% difference) but with clear intensification during 40 years. For the sake of fairness, we further removed the 3% difference in rainfall amount by decreasing this amount for 2010 rainfall timeseries (i.e., updated 2010 rainfall became 825mm for upstream and 1605 mm for downstream; Table S2). Keeping all input data between the two years the same except for rainfall, the simulated hydrological responses showed annual amounts of soil water, water yield, surface runoff, and baseflow for the upstream part of the WRB all

297 to become less under increased rainfall intensities (by -0.5%, -6%, -7%, and -8%,
respectively) whereas the responses increased for the downstream part (by 0.9%, 10%,
15%, and 3%, respectively; Table S2). Further, seasonal peak streamflow at the height
of the rainy season (June–August) increased by 51% and 15% for the up- and
downstream parts, respectively, whereas minimum dry-season flows (December–
February) became either less (-2% for the upstream part) or increased (by 25%
downstream; Table S2).

As we described in **Step 2**, the divergent soil water trends in Highland (Drying) and
Lowland (Wetting) to the intensified rainfall is universal. For a higher level scale (e.g.,
subbasin or basin), the Highland Drying Effect would be more obvious in the upstream
of a basin because of its greater portion of highland, and the Lowland Wetting Effect
would be more obvious in the downstream of a basin due to its greater portion of
lowland. That is why the upstream became dry and downstream became wet in the
WRB. In short, these model results can provide strong support for the contention that
the intensification of rainfall in the WRB is the chief factor underlying the observed
spatially diverging trends for water availability in recent decades, mainly as a result of
contrasts in the interaction between rainfall intensity, slope gradient and slope
morphology (Fig. S5).

**In summary**, rainfall intensification in the broad sense of the term can explain the
spatiotemporal pattern of the associated hydrological responses identified by the
process modeling, as also demonstrated convincingly by the results of rainfall intensity
scenario analyses across a range of spatial scales (Figs. S4 and S5; Table S2).

**B.** I am also not fully convinced with the robustness of the results, as they are based on
simulations from only one model. Although the model is calibrated against streamflow
from three gauges reasonably well (one might however question the model performance
at Tiane, where the simulated minimum flow is notably lower than observation in
validation period), no comparison is made to evaluate if the model can also reproduce
other parts of the hydrological cycles (evapotranspiration, soil moisture, water balance
etc.) reasonably well. There are many remote-sensing based products for such

evaluation.

**[Response]** We thank for the good comment and suggestion.

About **the SWAT model**, the first author of this article has relatively deep
understanding of the model structure/code and developed/published several modified
versions (e.g., R-SWAT-FME, SWAT-CO2, SWAT-DCDam, and SWAT-DayCent) (Wu
et al., 2012, 2014, 2016; Sun & Wu, 2022). Based on the numbers of model users and
relevant publications in the world, SWAT can be regarded as the most widely-used,
physically-based, daily timestep hydrological model at watershed scale, and thus it is a
key tool in the field of watershed hydrological processes and responses to climate
change and human activities (Arnold et al., 1998, 2012; Neitsch et al., 2011) in which
other popular models (VIC, MIKE, Xinanjiang, etc.) seem not comparable due to
various issues (e.g., Physical processes, Theoretical/Technical documentations, Open
source, Easy access/use, Timestep). Our experience and literature review could tell
SWAT is qualified for the study.

About the **model performance at the Tiane Station**, we agree that the original
streamflow simulation at this station did not perform well due to the clear under-
estimation of low flow. Therefore, we have **re-calibrated** the model parameters for up-,
mid-, and downstream, respectively, and thus we have **re-done** all the
simulations/analyses and **replotted** all figures/tables in our revision. As shown in new
Figure S6, the streamflow simulation has been improved.

About the **model validation**, following the reviewer's suggestion, we have added
remotely-sensed ET and soil water content to further validate the model in our revision.
For **ET validation**, as shown in Figure S7 (see **below**), we compared SWAT simulation
with re-analysis ET product based on a fusion of ERA5, MERRA2, and GLDAX2-
Noah data (Lu et al., 2021). About remotely-sensed **soil water**, there are substantial
differences in magnitude between products due to the great uncertainty of soil water
inversion and different soil depths those products used (Bitar et al., 2013; Martens et
al., 2017). For **soil water validation**, as did in previous publications, we focused on the
correlation between model and data (instead of magnitude). As shown in Figure S8 (see
**below**), we compared SWAT simulated soil water and the root zone soil water amount

from Centre Aval de Traitement des Données SMOS (CATDS) during 2011–2018 and
GLEAM-based topsoil water during 1980–2018.

In brief, **Figures S6, S7** and **S8** showed the model performed quite well in
simulations of streamflow, ET and soil water, and this revision makes our model
simulation more robust (please see **P8 L368-375**). We thank the reviewer for his/her
good suggestion.

**Figure S7.** Monthly comparison of SWAT simulated ET and re-analysis ET data based
on a fusion of ERA5, MERRA2, and GLDAX2-Noah data during 1980–
2017.

**Figure S8.** Annual comparison of SWAT simulated soil water and remotely-sensed root
zone soil water amount from Centre Aval de Traitement des Données
SMOS (CATDS) during 2011–2018 (a) and GLEAM-based topsoil water
during 1980–2018 (b).

Additionally, parts of the methodology and results interpretations need better
explanation:

- 1. What is the link between HRU and subbasin in the simulations, or how is simulated
results from HRUs upscaled to subbasins?

**[Response]** In a SWAT setup, a river basin (i.e., a study area) is delineated to a
number of subbasins based on topography, and a subbasin is divided into multiple
Hydrological Response Units (HRUs) based on unique combinations of land use,
soil type, and slope in the subbasin (Arnold et al., 1998; Neitsch et al., 2011). In this
study, the WRB was delineated into 187 subbasins and 2,738 HRUs. The least
computation unit of the model is HRU, then simulated hydrological components
(e.g., water yield, ET, soil water, surface runoff, baseflow) of all HRUs within a
subbasin were integrated to give the corresponding results per subbasin. A similar
integrative approach was used for the larger scales (i.e., the upstream, mid- and
downstream and the entire WRB). We have clarified this point in the Methodology
Section (please see P12 L559-562; L568-573).

2. What is the optimization criteria in calibration, and are the calibrated parameters
fixed for the whole region?

**[Response]** We adopted the widely-used SWAT-CUP for model calibration which
provides users with a few types of objective function (e.g., Square Error, R^2 , NSE),
and we selected NSE as the objective function. Then the maximization of NSE is the
optimization criteria in our model calibration. We selected five key parameters for
calibration for the up-, mid-, and downstream parts, respectively, and the resulting
fitted parameter values are listed in Table S3.

3. Would the EOF analysis leave out important patterns when the cumulative explained
variance is below 51% for the first two EOFs?

**[Response]** According to the North significance test, the first three modes were
statistically significant because the error range of the 4th mode overlapped with the
5th mode (Table S1). The explained variances by the first three modes are 35.0%,
14.6%, and 7.9%. The cumulative explained variance by the first two modes is
49.6%, and adding the 3rd mode (7.9%) would not improve the cumulative explained
variance significantly. Therefore, this way tells the first two modes could explain the
main spatiotemporal features. In addition, a previous study (Characteristics and
trends in various forms of the Palmer Drought Severity Index during 1900–2008,
*Journal of Geophysical Research: Atmospheres*, Aiguo Dai, 2011) adopted the first

two modes with a cumulative explained variance of 10%. Thus, our analysis and
result can be reliable.

4. Are differences in hydrological response between 1970 and 2010 due to rainfall
intensification alone? Are there any dams or other human driven changes between
1970 and 2010 in this basin?

**[Response]** Please see our responses to **Question A** by this reviewer (or see **P7**
**L333-341** or **below**):

To examine the hydrological responses of upstream and downstream of the West
River Basin (WRB) to the intensified rainfall, we designed two scenarios (1970 vs.
2010) to isolate the hydrological effects of rainfall intensity change alone. In the
scenario design, we selected two years with closely comparable rainfall totals but
strongly different intensities: **1970** (upstream rainfall and RI: 825 mm and 6.2;
downstream rainfall and RI: 1,605 mm and 9) and **2010** (upstream rainfall and RI: 849
420 mm and 6.5; downstream rainfall 1,650 mm and 11) because the amounts of rainfall in
these two years are very close (<3% difference) but with clear intensification during 40
422 years. For the sake of fairness, we further removed the 3% difference in rainfall amount
by decreasing this amount for 2010 rainfall timeseries (i.e., updated 2010 rainfall
became 825mm for upstream and 1605 mm for downstream; Table S2). That is to say,
all the input data for the hydrological simulation in the WRB are the same in the two
scenarios except for rainfall. Therefore, no other elements like human activities could
influence comparison results based on the scenario design.

5. Why it is concluded that wetting would occur downstream (instead of over the full
basin), when the change in discharge is much more drastic (>13 times) upstream?
Or are the two years not representative of the changing rainfall intensity in this basin?

**[Response]** Please see our responses to **Question A** and **Question 4** by this reviewer
where we explained how intensified rainfall caused spatially divergent hydrological
changes. Also, we have **re-calibrated** the model, **re-done** all simulations/analyses
and **replotted** all figures/tables in our revision. Thus, Table S2 has been updated.

6. I would leave out the minor issues so far, except for pointing out that in Table S2,
data in the last row is incorrect.

**[Response]** We thank for the good catch. To improve the model performance in
Tiane Station, as described in our responses to **Question B** by this Reviewer, we
have **re-calibrated** the model parameters and thus have **re-done** all the
simulations/analyses and **replotted** all figures/tables in our revision. Therefore,
simulation results including Table S2 have been updated accordingly in this revised
version.

**References**

- Arnold JG, Kiniry J, Srinivasan R, Williams J, Haney E, Neitsch S (2012) Soil and water assessment tool
input/output documentation version 2012. *Texas water resources institute*, **7**.
- Arnold JG, Srinivasan R, Muttiah RS, Williams JR (1998) Large area hydrologic modeling and
assessment part i: Model development. *J. Am. Water Resources Association*, **34**, 73-89.
- Berghuijs WR, Harrigan S, Molnar P, Slater LJ, Kirchner JW (2019) The relative importance of different
flood-generating mechanisms across europe. *Water Resources Research*, **55**, 4582-4593.
- Berghuijs WR, Slater LJ (2023) Groundwater shapes north american river floods. *Environmental*
*Research Letters*, **18**, 034043.
- Berghuijs WR, Woods RA, Hutton CJ, Sivapalan M (2016) Dominant flood generating mechanisms
across the united states. *Geophysical Research Letters*, **43**, 4382-4390.
- Bibi S, Song Q, Zhang Y, Liu Y, Kamran MA, Sha L, *et al.* (2021) Effects of climate change on terrestrial
water storage and basin discharge in the lancang river basin. *Journal of Hydrology: Regional*
*Studies*, **37**, 100896.
- Bitar AA, Kerr Y, Wigneron MOCF (2013) In *IEEE International Geoscience and Remote Sensing*
*Symposium, IGARSS 2013*.
- Chou C, Chiang JCH, Lan C-W, Chung C-H, Liao Y-C, Lee C-J (2013) Increase in the range between
wet and dry season precipitation. *Nature Geoscience*, **6**, 263-267.
- Han S-C, Ghobadi-Far K, Yeo I-Y, McCullough CM, Lee E, Sauber J (2021) Grace follow-on revealed
bangladesh was flooded early in the 2020 monsoon season due to premature soil saturation.
*Proceedings of the National Academy of Sciences*, **118**, e2109086118.
- Kendall KA, Shanley JB, McDonnell JJ (1999) A hydrometric and geochemical approach to test the
transmissivity feedback hypothesis during snowmelt. *Journal of Hydrology*, **219**, 188-205.
- Li J, Wu C, Xia C-A, Yeh PJF, Hu BX, Huang G (2021) Assessing the responses of hydrological drought
to meteorological drought in the huai river basin, china. *Theoretical and Applied Climatology*,
**144**, 1043-1057.
- Liu B, Xu M, Henderson M, Qi Y (2005) Observed trends of precipitation amount, frequency, and
intensity in china, 1960–2000. *Journal of Geophysical Research: Atmospheres*, **110**.
- Lu J, Wang G, Chen T, Li S, Hagan DFT, Kattell G, *et al.* (2021) A harmonized global land evaporation
dataset from model-based products covering 1980–2017. *Earth System Science Data*, **13**, 5879-
5898.
- Mahmood S, Rahman A-u (2019) Flash flood susceptibility modeling using geo-morphometric and
hydrological approaches in panjkora basin, eastern hindu kush, pakistan. *Environmental Earth*

*Sciences*, **78**, 43.

Martens B, Miralles DG, Lievens H, Van Der Schalie R, De Jeu RA, Fernández-Prieto D, *et al.* (2017)

Gleam v3: Satellite-based land evaporation and root-zone soil moisture. *Geoscientific Model*

*Development*, **10**, 1903-1925.

Mishra A, Liu SC (2014) Changes in precipitation pattern and risk of drought over india in the context

of global warming. *Journal of Geophysical Research: Atmospheres*, **119**, 7833-7841.

Neitsch SL, Arnold JG, Kiniry JR, Williams JR (2011) Soil and water assessment tool theoretical

documentation version 2009. Texas Water Resources Institute.

Sharma A, Wasko C, Lettenmaier DP (2018) If precipitation extremes are increasing, why aren't floods?

*Water Resources Research*, **54**, 8545-8551.

Shiu C-J, Liu SC, Fu C, Dai A, Sun Y (2012) How much do precipitation extremes change in a warming

climate? *Geophysical Research Letters*, **39**.

Sun P, Wu Y (2022) Dynamic modeling framework of sediment trapped by check-dam networks: A case

study of a typical watershed on the chinese loess plateau. *Engineering*.

Wang Y, Wang L, Zhou J, Yao T, Yang W, Zhong X, *et al.* (2021) Vanishing glaciers at southeast tibetan

plateau have not offset the declining runoff at yarlung zangbo. *Geophysical Research Letters*,

**48**, e2021GL094651.

Wasko C, Nathan R (2019) Influence of changes in rainfall and soil moisture on trends in flooding.

*Journal of Hydrology*, **575**, 432-441.

Wasko C, Nathan R, Peel MC (2020) Changes in antecedent soil moisture modulate flood seasonality in

a changing climate. *Water Resources Research*, **56**, e2019WR026300.

Wasko C, Shao Y, Vogel E, Wilson L, Wang QJ, Frost A, *et al.* (2021) Understanding trends in hydrologic

extremes across australia. *Journal of Hydrology*, **593**, 125877.

Wu Y, Chen J (2012) An operation-based scheme for a multiyear and multipurpose reservoir to enhance

macroscale hydrologic models. *Journal of Hydrometeorology*, **13**, 270-283.

Wu Y, Liu S (2014) Improvement of the r-swat-fme framework to support multiple variables and multi-

objective functions. *Science of The Total Environment*, **466**, 455-466.

Wu Y, Liu S, Abdul-Aziz OI (2012) Hydrological effects of the increased co2 and climate change in the

upper mississippi river basin using a modified swat. *Climatic Change*, **110**, 977-1003.

Wu Y, Liu S, Qiu L, Sun Y (2016) Swat-daycent coupler: An integration tool for simultaneous hydro-

biogeochemical modeling using swat and daycent. *Environmental modelling & software*, **86**,

81-90.

Xu F, Zhou Y, Zhao L (2022) Spatial and temporal variability in extreme precipitation in the pearl river

basin, china from 1960 to 2018. *International Journal of Climatology*, **42**, 797-816.

Yang L, Yang Y, Villarini G, Li X, Hu H, Wang L, *et al.* (2021) Climate more important for chinese flood

changes than reservoirs and land use. *Geophysical Research Letters*, **48**.

Zhou G, Wei X, Wu Y, Liu S, Huang Y, Yan J, *et al.* (2011) Quantifying the hydrological responses to

climate change in an intact forested small watershed in southern china. *Global Change Biology*,

**17**, 3736-3746.

REVIEWERS' COMMENTS

Reviewer #1 (Remarks to the Author):

The authors addressed most of my concerns in the previous review. However, I still recommend the author to remove the 9-dash line in the sea map in the inset plot of Figure 1 to avoid controversy. This sea territory is a politically controversial topic and not necessary for this work. The land map of China is good enough to locate the study area.

Then, I think the manuscript is good for publication.

Reviewer #3 (Remarks to the Author):

General Feedback:

Firstly, I commend the authors for their dedicated efforts in addressing reviewers' comments from the previous round. Notably, the authors have augmented their work by incorporating detailed explanations elucidating the mechanisms underlying the spatially divergent hydrological response to escalating rainfall intensities—a focal point of this paper. Significant enhancements have been achieved in refining the model's performance and evaluation, accompanied by corrections in the case studies following the authors' re-analysis. The phenomenon investigated in this paper holds broad interest, justifying its publication in Nature Communications. However, some additional clarifications are essential before the acceptance of the paper.

Specific Comments:

Rationale and Simulation Results (Question A):

- The rationale in response to Question A is generally coherent. However, please consider providing additional elucidation in Step 2, which predominantly presents simulation results without thorough explanation. It would be beneficial to understand why a higher slope results in drying under intensified rainfall. Does the SWAT model simulate an accelerated water flow from the Hydrological Response Unit (HRU)? Is there a greater proportion of water exiting upstream and reaching the downstream region? Additionally, could rising temperatures contribute to the trends observed in Figure S5? Please specify whether the simulations, keeping all variables constant except rainfall, employed the 1970 or 2010 temperature.

Inclusion of Maps (Figure 4):

- Given the pivotal role of slope in hydrological responses, it would be highly informative for readers if you could please consider including maps depicting average slope at the subbasin and HRU levels in Figure 4.

Minor Comments:

- Inconsistent Terminology: To ensure clarity, please strive for consistency in the use of terminology. For instance, the basin was delineated into 187 subbasins and 2738 HRUs (P12L569). However, when

referring to "Subbasin scale (upstream/downstream)" (P7L333), it's unclear whether this encompasses the entire upstream and downstream regions of the basin or just two of the 187 subbasins. Likewise, in Figure S5, which is purportedly analyzed at the subbasin level (SI, P4L45), the drawn polygons appear to be at the HRU level. Could you please consistently use terminology such as "sub-region" (SI, P2L26) for better clarity?

- P4L154-159: Please consider clarifying that the decrease in the number of light rain days alone does not necessarily indicate greater drying. Instead, it can be inferred from increases in the number of rainless days.
- P5L219-220: Could you please consider modifying to "This result aligns with the spatial hydrological response patterns identified in the previous section, indicating drought..."
- P5L228: To enhance the comprehensiveness of your work, please consider including a reference for the North significance test.
- P11L512: Might I suggest changing to "The West River is the second-largest in China in terms of streamflow, with an average..."

Response to Reviewer's Comments

Reviewer 1

The authors addressed most of my concerns in the previous review. However, I still recommend the author to remove the 9-dash line in the sea map in the inset plot of Figure 1 to avoid controversy. This sea territory is a politically controversial topic and not necessary for this work. The land map of China is good enough to locate the study area.

Then, I think the manuscript is good for publication.

[Response] This study is about climate change and hydrological science, and the map is merely meant to indicate the location of the study basin which is in the East Asian monsoon region. Thus, a terrestrial map can help serve the purpose.

Reviewer 3

General Feedback:

Firstly, I commend the authors for their dedicated efforts in addressing reviewers' comments from the previous round. Notably, the authors have augmented their work by incorporating detailed explanations elucidating the mechanisms underlying the spatially divergent hydrological response to escalating rainfall intensities—a focal point of this paper. Significant enhancements have been achieved in refining the model's performance and evaluation, accompanied by corrections in the case studies following the authors' re-analysis. The phenomenon investigated in this paper holds broad interest, justifying its publication in Nature Communications. However, some additional clarifications are essential before the acceptance of the paper.

[Response] We thank the Reviewer for his/her very positive evaluation of our work, and we have made clarifications required in details as shown below.

Specific Comments:

Rationale and Simulation Results (Question A):

- The rationale in response to Question A is generally coherent. However, please

consider providing additional elucidation in Step 2, which predominantly presents simulation results without thorough explanation. It would be beneficial to understand why a higher slope results in drying under intensified rainfall. Does the SWAT model simulate an accelerated water flow from the Hydrological Response Unit (HRU)? Is there a greater proportion of water exiting upstream and reaching the downstream region? Additionally, could rising temperatures contribute to the trends observed in Figure S5? Please specify whether the simulations, keeping all variables constant except rainfall, employed the 1970 or 2010 temperature.

[Response]

SWAT adopts an equation (Williams, 1995) to adjust the SCS Curve Number (CN) to a different slope. The equation tells that the higher the slope, the greater the CN will be, suggesting more overland flow and less infiltration with a higher slope when other conditions remain unchanged. Thus, SWAT can simulate an increase in overland flow (but decreased subsurface flow) from the HRU (as shown in Figure S4). About Figure S5, we take the specific small watershed in the upstream (the left panel of the figure) as an example to explain this concern. **There is only one weather station for this specific small watershed, then the climate conditions (e.g., precipitation and temperature) are the same for all the slopes in the watershed.** The trends in temperature (e.g., warming) would not influence the relationship between Slope and Soil Water Trend because such a warming temperature is the same for all the slopes. The simulations as described in Step 3 (Table S2) keep all variables the same (i.e., using 1970 temperature) for two scenarios except for rainfall—1970 rainfall and 2010 rainfall are used for the two scenarios, respectively. Because 2010 rainfall is about 3% more than 1970 rainfall in amount, for the sake of fairness, we removed this difference for 2010 rainfall timeseries (i.e., updated 2010 rainfall became 825mm for upstream and 1605 mm for downstream; Table S2).

The above clarifications have been reflected in this revision. Please see P12 L556-558, P7 L319-321, and P7 L335-336.

Inclusion of Maps (Figure 4):

- Given the pivotal role of slope in hydrological responses, it would be highly

informative for readers if you could please consider including maps depicting average slope at the subbasin and HRU levels in Figure 4.

[Response] We thank for the good suggestion. We have revised Figure 4 as suggested (Please see below).

Figure 4. Spatial distribution of annual trends for four hydrological components from 1965 to 2018 and slope across the West River Basin. **a, c, e, and g** The annual trends for soil water content, water yield, surface runoff, and baseflow at subbasin level (shaded areas with statistically significant trends). **b, d, f, and h** The annual trends for soil water content, water yield, surface runoff, and baseflow at hydrological response unit (HRU) level. **i, j** Slopes for subbasin and HRU level, respectively.

Minor Comments:

- Inconsistent Terminology: To ensure clarity, please strive for consistency in the use of

terminology. For instance, the basin was delineated into 187 subbasins and 2738 HRUs (P12L569). However, when referring to "Subbasin scale (upstream/downstream)" (P7L333), it's unclear whether this encompasses the entire upstream and downstream regions of the basin or just two of the 187 subbasins. Likewise, in Figure S5, which is purportedly analyzed at the subbasin level (SI, P4L45), the drawn polygons appear to be at the HRU level. Could you please consistently use terminology such as "sub-region" (SI, P2L26) for better clarity?

[Response] Following this Reviewer's suggestion, we used 'sub-region' to refer to 'entire upstream or downstream' to avoid misunderstanding (see P7 L326) as we do in the Supplementary Information. Also, we have corrected that the relationship between Slope and Soil Water Trend is drawn at the HRU level (see Figure S5).

- P4L154-159: Please consider clarifying that the decrease in the number of light rain days alone does not necessarily indicate greater drying. Instead, it can be inferred from increases in the number of rainless days.

[Response] Done as suggested (please see P3 L144-147).

- P5L219-220: Could you please consider modifying to "This result aligns with the spatial hydrological response patterns identified in the previous section, indicating drought..."

[Response] Done (please see P5 L207-210).

- P5L228: To enhance the comprehensiveness of your work, please consider including a reference for the North significance test.

[Response] Done (please see P5 L216-218).

- P11L512: Might I suggest changing to "The West River is the second-largest in China in terms of streamflow, with an average..."

[Response] Done (please see P11 L503-506).

References

Williams J (1995) Chapter 25: The epic model. Computer models of watershed hydrology, 1, 909-1000.